# Context-Aware Entity Grounding with Open-Vocabulary 3D Scene Graphs

**Haonan Chang[1] , Kowndinya Boyalakuntla[1], Shiyang Lu[1], Siwei Cai[2], Eric Pu Jing[1], Shreesh Keskar[1]**

**Shijie Geng[1], Adeeb Abbas[2], Lifeng Zhou[2], Kostas Bekris[1], Abdeslam Boularias[1]**

[1]`hc856,kb1204,sl1642,epj25,skk139,sg1309,kb572,ab1544@scarletmail.rutgers.edu`
[2]`sc3568@drexel.edu,adeebabbs@gmail.com,lz457@drexel.edu`
[1]`Rutgers University` [2]`Drexel University`

**Abstract:** We present an **O**pen-**V**ocabulary 3D **S**cene **G**raph (OVSG), a formal framework for grounding a variety of entities, such as object instances, agents, and regions, with free-form text-based queries. Unlike conventional semantic-based object localization approaches, our system facilitates context-aware entity localization, allowing for queries such as "pick up a cup on a kitchen table" or "navigate to a sofa on which someone is sitting". In contrast to existing research on 3D scene graphs, OVSG supports free-form text input and open-vocabulary querying. Through a series of comparative experiments using the ScanNet [1] dataset and a self-collected dataset, we demonstrate that our proposed approach significantly surpasses the performance of previous semantic-based localization techniques. Moreover, we highlight the practical application of OVSG in real-world robot navigation and manipulation experiments. The code and dataset used for evaluation can be found at https://github.com/changhaonan/OVSG.

**Keywords:** Open-Vocabulary Semantics, Scene Graph, Object Grounding

## 1 Introduction

In this paper, we aim to address a fundamental problem in robotics – grounding semantic entities within the real world. Specifically, we explore how to unambiguously and accurately associate entities present in commands, such as object manipulation, navigation to a specific location, or communication with a particular user.

Currently, the prevailing method for grounding entities in the robotics domain is semantic detection [2]. Semantic detection methods are intuitive and stable. However, in scenes with multiple entities of the same category, semantic labels alone cannot provide a unique specification. In contrast, humans naturally possess the ability to overcome this grounding ambiguity by providing context-aware specifications, such as detailed descriptions and relative relations. For example, rather than simply designating "a cup", humans often specify "a blue cup on the shelf", "a coffee cup in the kitchen", or "Mary's favorite tea cup".

Inspired by this, a series of recent works introduce context relationship into grounding problem [3, 4, 5, 6, 7]. These approaches employ 3D scene graphs as a scene representation that concurrently accounts for instance categories and inter-instance spatial contexts. In a 3D scene graph, concepts such as people, objects, and rooms are depicted as nodes, with attributes like color, material, and affordance assigned as node attributes. Moreover, spatial relationships are represented as graph edges. Such structure enables 3D scene graphs to seamlessly support context-aware object queries,

such as "the red cup on the table in the dining room", provided that the attribute, the semantic category, and the relationship have been predefined in the graph.

However, this inevitably brings us to a more crucial question that this paper aims to answer: how do we cope with scenarios when the class category, relationship, and attribute are not available in the constructed 3D scene graph? Tackling this question is vital if we wish to effectively integrate robots into real-world scenarios. To resolve the challenge, we present a novel framework in this paper – the Open-Vocabulary 3D Scene Graph (OVSG). To the best of our knowledge, OVSG is the first 3D scene graph representation that facilitates context-aware entity grounding, even with unseen semantic categories and relationships.

To evaluate the performance of our proposed system, we conduct a series of query experiments on ScanNet [1], ICL-NUIM [8], and a self-collected dataset DOVE-G (**D**ataset for **O**pen-**V**ocabulary **E**ntity **G**rounding). We demonstrate that by combining open-vocabulary detection with 3D scene graphs, we can ground entities more accurately in real-world scenarios than using the state-of-the-art open-vocabulary semantic localization method alone. Additionally, we designed two experiments to investigate the open-vocabulary capability of our framework. Finally, we showcase potential applications of OVSG through demonstrations of real-world robot navigation and manipulation.

Our contributions are threefold: 1) A new dataset containing eight unique scenarios and 4,000 language queries for context-aware entity grounding. 2) A novel 3D scene graph-based method to address the context-aware entity grounding from open-vocabulary queries. 3) Demonstrate the real-world applications of OVSG, such as context-aware object navigation and manipulation.

## 2  Related Work

**Open-Vocabulary Semantic Detection and Segmentation**  The development of foundation vision-language pre-trained models, such as CLIP [9], ALIGN [10], and LiT [11], has facilitated the progress of 2D open-vocabulary object detection and segmentation techniques [12, 13, 14, 15, 16, 17, 18]. Among these approaches, Detic [16] stands out by providing open-vocabulary instance-level detection and segmentation simultaneously. However, even state-of-the-art single-frame methods like Detic suffer from perception inconsistency due to factors such as view angle, image quality, and motion blur. To address these limitations, Lu et al. proposed OVIR-3D [19], a method that fuses the detection result from Detic into an existing 3D model using 3D global data association. After fusion, the 3D scene is segmented into multiple instances, each with a unique Detic feature attached. Owing to its stable performance, we choose OVIR-3D as our semantic backbone.

**Vision Language Object Grounding**  In contrast with object detection and segmentation, object grounding focuses on pinpointing an object within a 2D image or a 3D scene based on textual input. In the realm of 2D grounding, various studies, such as [20, 21, 22, 23], leverage vision-language alignment techniques to correlate visual and linguistic features. In the 3D context, object grounding is inherently linked to the challenges of robot navigation, thus gaining significant attention from the robotics community. For instance, CoWs [24] integrates a CLIP gradient detector with a navigation policy for effective zero-shot object grounding. More recently, NLMap [25], ConceptFusion [26], CLIP-Fields [27] opts to incorporate pixel-level open-vocabulary features into a 3D scene reconstruction, resulting in a queryable scene representation. While NLMap overlooks intricate relationships in their framework, ConceptFusion claims to be able query objects from long text input with understanding of the object context. Thus, we include ConceptFusion as one of our baselines for 3D vision-language grounding.

**3D Scene Graph**  3D scene graphs provide an elegant representation of objects and their relationships, encapsulating them as nodes and edges, respectively. The term "3D" denotes that each node within the scene possesses a three-dimensional position. Such graph structure has been widely researched for decades in robotics community [3, 4, 5, 7, 6, 28, 29],. In [3], Fisher et al. first introduced the concept of 3D scene graphs, where graph nodes are categorized by geometric shapes. Armeni et al. [4] and Kim et al. [5] then revisited this idea by incorporating semantic labels to graph

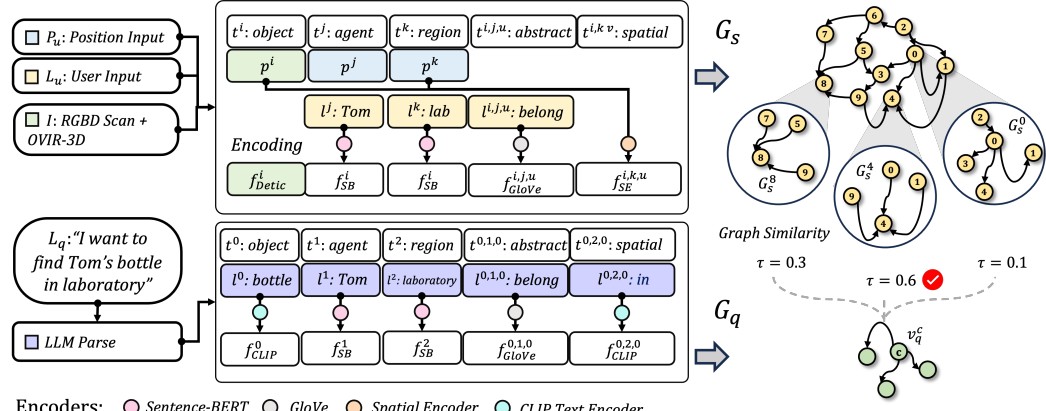

Figure 1: This is an illustration of the proposed pipeline. The system inputs are the positional input $P_u$, user input $L_u$, RGBD Scan $I$, and a query language $L_q$. The top section depicts the construction of $G_s$. Both $P_u$ and $L_u$ are directly fed into $G_s$. The RGBD Scan $I$ inputs into the open-vocabulary fusion system referred to as *OVIR-3D*. This system outputs a position and a Detic feature for each object. Subsequently, the language descriptions for the agent and region are converted into features via different encoders. A unique Spatial Relationship Encoder is employed to encode spatial relationship features from pose pairs. The bottom section shows the building of $G_q$. The query $L_q$ used in this example is, "I want to find Tom's bottle in the laboratory." An LLM is used to parse it into various elements, each with a description and type. These descriptions are then encoded into features by different encoders, forming $G_q$. Finally, grounding the query language $L_q$ within scene $S$ becomes a problem of locating $G_q$ within $G_s$. A candidate proposal and ranking algorithm are introduced for this purpose. The entity we wish to locate is represented by the central node of the selected candidate.

nodes. These works establish a good foundation for semantic-aware 3D scene graphs, demonstrating that objects, rooms, and buildings can be effectively represented as graph nodes. Recently, Wald et al. [7] showed that 3D feature extraction and graph neural networks (GNN) can directly infer semantic categories and object relationships from raw 3D point clouds. Rosinol et al. [6] further included dynamic entities, such as users, within the scope of 3D scene graph representation. While 3D scene graphs exhibit great potential in object retrieval and long-term motion planning, none of the existing methods support open-vocabulary queries and direct natural language interaction. Addressing these limitations is crucial for real-world deployment, especially for enabling seamless interaction with users.

## 3 Open-Vocabulary 3D Scene Graph

### 3.1 Open-Vocabulary 3D Scene Graph Representation

An Open-Vocabulary 3D Scene Graph (OVSG) is denoted as $G = |V, E|$, where $V$ signifies graph nodes and $E$ stands for graph edges. Each node $v^i$ in $V = \{v^i\} = \{t^i, f^i, l^i, p^i\}$ consists of a node type $t^i$, a open-vocabulary feature $f^i$, a language description $l^i$ (optional), and a 3D position $p^i$ (optional); $i$ is the node index. In this study, we identify three primary node types, $t^i$: object, agent, and region. The open-vocabulary feature $f^i$ associated with each node $v_i$ is contingent on the node type $t_i$. The encoder utilized for $f^i$ is accordingly dependent on $t_i$. The 3D position $p^i = \{x_c, y_c, z_c, x_{min}, y_{min}, z_{min}, x_{max}, y_{max}, z_{max}\}$ of each entity is defined by a 3D bounding box and its center position. Edges in the graph are represented by $E = \{e^{i,j} | v^i, v^j \in V\}, e^{i,j} = \{r^{i,j,k} = \{t^{i,j,k}, f^{i,j,k}, l^{i,j,k}\} | k = 0, \ldots\}$. Each edge $e^{i,j}$ encapsulates all relationships $r^{i,j,k}$ between the nodes $v^i$ and $v^j$. The triplet notation $(i, j, k)$ refers the $k^{th}$ relationship between node $v^i$ and $v^j$, $t^{i,j,k}$ indicates the type of this relationship. We primarily categorize two relationships in this study: spatial relations and abstract relationships. A short sentence $l^i$ is optionally provided to describe this relationship. The feature $f^{i,j,k}$ encodes the semantic meaning of the relationship, whose encoder depends on $t^{i,j,k}$. For a more detailed definition of these types, please refer to Section 3.3.

The primary distinction of OVSG from conventional 3D scene graph work is its utilization of semantic features, instead of discrete labels, to characterize nodes and relationships. These features are either directly trained within the language domain like Sentence-BERT [30] and GloVe [31], or aligned to it, as seen with CLIP [9] and Detic [16]. The versatility of language features enables

OVSG to handle diverse queries. The degree of similarity among nodes and edges is depicted using a distance metric applied to their features:

$$\text{dist}(v^i, v^j) = \begin{cases} \infty & \text{if } t^i \neq t^j \\ 1 - \text{dot}(f^i, f^j) & \text{else} \end{cases} \;;\; \text{dist}(e^{i,j}, e^{u,v}) = \min_{\forall k \in |e^{i,j}|, \forall w \in |e^{u,v}|} \text{dist}(r^{i,j,k}, r^{u,v,w})$$

$$\text{dist}(r^{i,j,k}, r^{u,v,w}) = \begin{cases} \infty & \text{if } t^{i,j,k} \neq t^{u,v,w} \\ 1 - \text{dot}(f^{i,j,k}, f^{u,v,w}) & \text{if } t^{i,j,k} = t^{u,v,w} \neq \text{spatial} \\ \text{SRP}(f^{i,j,k}, f^{u,v,w}) & \text{if } t^{i,j,k} = t^{u,v,w} = \text{spatial} \end{cases} \qquad (1)$$

, where the $|e^{i,j}|$ and $|e^{u,v}|$ are the number of relationships inside $e^{i,j}$ and $e^{u,v}$; *SRP* refers to a Spatial Relationship Predictor. Check Section 3.3 and Appendix B for more details. Noticeably, the distance across different types will not be directly compared. These distances will be used to compute the type-free index in Section 3.4.

### 3.2 Context-Aware Open-Vocabulary Entity Grounding

The problem we address can be formally defined using the open-vocabulary scene graph concept as follows: Given a scene, represented as $S$, our objective is to localize an entity, referred to as $s$, using natural language, represented as $L_q$, within the context of the scene $S$. Essentially, we seek to establish a mapping $\pi$ such that $s = \pi(L_q|S)$. An RGBD scan of the scene $I$, user linguistic input $L_u$, and position input $P_u$ are provided to facilitate this process. Significantly, the query language $L_q$ may encompass entity types and relationship descriptions not previously included in the scene graph construction phase.

Our proposed procedure can be separated into two main stages. The first stage involves the construction of the scene graph. From the user input $L_u$ and the RGBD scan $I$, we construct an open-vocabulary scene graph (OVSG) for the entire scene, denoted as $G_s$. This is a one-time process that can be reused for every subsequent query. When a new query is introduced, we also construct an OVSG using the query $L_q$, denoted as $G_q$. Once we have both scene graphs $G_s$ and $G_q$, we proceed to the second stage, which is the graph matching stage. Here, we match the query scene graph, $G_q$, with a sub-graph from the whole scene graph, $G_s$. The queried entity is situated within the matched sub-graph.

### 3.3 3D Scene Graph Building

**Type definition** Prior to delving into the scene graph construction procedure, we first delineate the categories of node types and edge types this paper pertains to. The term *Object* signifies static elements within a scene, such as sofas, tables, and so forth. The term *Agent* is attributed to dynamic, interactive entities in the scene, which could range from humans to robots. *Region* indicates a specific area, varying in scale from the surface of a tabletop to an entire room or building. Regarding relationships, *spatial* describes positional relationships between two entities, such as Tom being in the kitchen. Conversely, *abstract* relationships are highly adaptable, enabling us to elucidate relationships between an agent and an object (for instance, a cup belonging to Mary) or the affordance relationship between two objects, such as a key being paired with a door.

**Input process** The inputs for $G_s$ consist of an RGBD-scan set $I$, a user language input $L_u$, and a user position input $P_u$. The $L_u$ input assigns names to agents and regions and provides descriptions of abstract relationships. $P_u$ provides the locations for the agent and region (not including object position), and it can be autonomously generated using existing algorithms like DSGS [6]. Since this process is not the focus of our study, we assume $P_u$ is pre-determined in this paper. The input $I$ is an RGBD scan of the entire scene, which is fed into the **O**pen-**V**ocabulary **3D** **I**nstance **R**etrieval (OVIR-3D) [19] system, a fusion system operating at the instance level. OVIR-3D returns a set of objects, each denoted by a position $p^i$ and a Detic feature $f^i_{Detic}$.

$G_q$ accepts a language query $L_q$ as its input. An exemplary query, as depicted in Figure 1, is "I want to find Tom's bottle in laboratory". To parse this language, we utilize a large language model (LLM),

such as GPT-3.5 or LLAMA. Utilizing a meticulously engineered prompt (refer to Appendix A for more details), we can interpret different entities within the query.

**Feature encoding** As specified in Eq. 1, the calculation of the similarity between nodes and edges relies heavily on their features. This operation of computing features is termed the feature encoding process.

Instead of using a unified encoder as in previous works [25, 26], we choose different encoders for various node and relationship types. Since the inputs of $G_s$ and $G_q$ differ, the selection of encoders for each graph also varies. Object features in $G_s$ are generated by deploying OVIR-3D to the 3D scan of the scene. These features are Detic features. Meanwhile, objects in $G_s$ are encoded from their names $l$ (parsed from LLM during the input process) using the CLIP-text encoder. Because the Detic feature is directly trained to align with the CLIP-text feature, we can compute distances for object nodes between $G_s$ and $G_q$ using Eq.1. For agent and region nodes in $G_s$, they are identified by their names in the user input, $L_u$. Whereas in $G_q$, agent and region nodes are also specified by names $l$. For both of them, we employ Sentence-BERT [30] to encode the language features. As for relationships, we differentiate between spatial relationships and abstract relationships. In $G_s$, the input for spatial relationships comes from the positions of the corresponding nodes. In contrast, in $G_q$, the input for spatial relationships comes from language descriptions $l$ parsed from $L_q$ by LLM. Given the absence of a standardized approach for spatial-language encoding, we trained a spatial encoder for this purpose (see Appendix B). Finally, for abstract relationship features in $G_s$, the input is language $l$ from user input, $L_u$. In $G_q$, the input is also textual. We use GloVe to encode these texts on both sides.

Multiple distinct encoders are utilized during the feature encoding step. Different encoders have varied emphases, and using a combination can improve the robustness of OVSG. For instance, GloVe is trained to be sensitive to nuances like sentiment, while Sentence-BERT is not. Therefore, we use GloVe for abstract relationships to better distinguish relationships such as "like" and "dislike". Conversely, while GloVe does have a predefined vocabulary list, Sentence-BERT does not. Hence, for encoding the names of agents and regions, we prefer Sentence-BERT. Moreover, OVSG is designed with a modularized structure, allowing future developers to easily introduce new types and feature encoders into OVSG.

### 3.4 Sub-graph Matching

Subsequent to the phases of input processing and feature encoding, two OVSG representations are constructed: one for the scene and another for the query, denoted by $G_s$ and $G_q$ respectively. The problem of grounding $L_q$ within the scene $S$ can be converted now effectively translates to locating $G_q$ within $G_s$. Generally, the subgraph-matching problem is NP-hard, prompting us to make several assumptions to simplify this problem. In this study, we assume that our $G_q$ is a star graph, signifying that a central node exists and all other nodes are exclusively linked to this central node. (If $G_q$ is not a star-graph, we will extract a sub-star-graph from it, and use this sub-graph as our query graph.)

The pipeline of sub-graph matching is illustrated on the right side of Figure 1. This a two-step procedure: candidate proposal and re-ranking. Let's denote the center of $G_q$ as $v_q^c$. Initially, we traverse all nodes, $v_s^i$, in $V_s$, ranking them based on their distance to $v_q^c$, computed with Eq. 1. Subsequently, we extract the local subgraph, $G_s^i$, surrounding each candidate, $v_s^i$. These extracted subgraphs serve as our candidate subgraphs. In the second phase, we re-rank these candidates using a graph-similarity metric, $\tau(G_q, G_s^i)$. To evaluate graph similarity, we examine three distinct methodologies: Likelihood, Jaccard coefficient, and Szymkiewicz-Simpson index.

**Likelihood** Assuming the features of nodes and edges all originate from a normal distribution, we can define the likelihood of nodes and edges being identical as follows: $L(v^i, v^j) = exp(\frac{-\text{dist}(f^i, f^j)}{\sigma_v})$ for nodes and $L(e^{i,j}, e^{u,v}) = exp(\frac{-\text{dist}(f^{i,j}, f^{u,v})}{\sigma_e})$ for edges. Here $\sigma_v$ and $\sigma_e$ are balancing parameters. From this, we can derive the graph-level likelihood $\tau_L$ as:

$$\tau_L(G_q, G_s^i) = L(v_q^c, v_{s^i}^c) \times \prod_{k \in |V_q|} \underset{j \in |V_{s^i}|}{\text{argmax}} [L(v_q^k, v^j) \cdot L(e_q^{c,k}, e_{s^i}^{c,j})] \qquad (2)$$

where $v_{s^i}^c$ is the center node of $G_s^i$. The insight behind this formula is to iterate over all possible node-level associations and select the one that maximizes the overall likelihood that $G_q$ matches with $G_s^i$. Noticeably, we use $\sigma_v$ and $\sigma_e$ to balance the node-wise and edge-wise likelihood. In practice, we use $\sigma_v = 1.0$ and $\sigma_e = 2.0$ to make the matching more sensitive to node-level semantics.

**Jaccard-coefficient & Szymkiewicz–Simpson index** In addition to the likelihood index, we also consider other widely used graph similarity indices such as the Jaccard and Szymkiewicz–Simpson indices. Both indices measure the similarity between two sets.

We adopt a similar method as in [7], generating a set $S(G)$ for each graph $G$ by combining nodes and edges, such that $|S(G)| = |V| + |E|$. The Jaccard coefficient $\tau_J$ and Szymkiewicz–Simpson index $\tau_S$ are then defined as follows:

$$\tau_J(G_q, G_s^i) = \frac{|S(G_q) \cap S(G_s^i)|}{|S(G_q)| + |S(G_s^i)| - |S(G_q) \cap S(G_s^i)|}, \tau_S(G_q, G_s^i) = \frac{|S(G_q) \cap S(G_s^i)|}{min(|S(G_q)|, |S(G_s^i)|)} \qquad (3)$$

Given that we already know $|S(G_q)|$ and $|G_s^i|$, we simply need to compute $|S(G_q) \cap S(G_s^i)|$, which consists of nodes or edges that belong to both $G_q$ and $G_s^i$. We can define this union by applying distance thresholds $\tau_v$ and $\tau_e$ for node and edge separately:

$$S(G_q) \cap S(G_s^i) = \{(v_q^k, v_{s^i}^{\pi(k)})|dist(f_q^k, f_{s^i}^{\pi(k)}) < \epsilon_v\} + \{(e_q^k, e_{s^i}^{\pi(k)})|dist(e_q^k, e_{s^i}^{\pi(k)}) < \epsilon_e\} \quad (4)$$

Here, $\pi$ is a data association between $G_q$ and $G_s^i$, where $\pi(k) = \text{argmin}_{\pi(k)}(dist(s_k, s_{\pi(k)}))$. $\epsilon_v$ and $\epsilon_e$ are threshold parameters. The differences between $\tau_L$, $\tau_J$, and $\tau_S$ can be understood as follows: $\tau_L$ describes the maximum likelihood among all possible matches between $G_q$ and $G_s^i$. Both $\tau_J$ and $\tau_S$ use thresholds $\epsilon_v$, $\epsilon_e$ to convert the node and edge matches to binary, and they measure the overall match rate with different normalization.

## 4 System Evaluation

Our OVSG framework experiments addressed these research questions: 1) How does our context-aware grounding method compare to prevailing approaches, including the SOTA semantic method and the recent work in the landscape of 3D semantic/spatial mapping, ConceptFusion [32] 2) How well does OVSG handle open-vocabulary queries? 3) What differences do our graph similarity-based methods show? 4) How well does OVSG perform inside a real robot environment?

These questions are imperative as they not only test the robustness of the OVSG framework but also its comparative efficacy against notable methods like ConceptFusion in the ability to handle the intricacies of context-aware open-vocabulary queries.

### 4.1 Queries, Dataset, Metrics & Baselines

**Queries** We have two categories of queries for evaluation:

- **Object-only Queries** These queries are devoid of any specific agent or region preference. They are less generic and assess the system's grounding ability based purely on objects. An example might be: "Can you identify a monitor with a keyboard positioned behind it?"

- **Whole Queries** These queries inherently contain a mix of agent, region, and object preferences. For instance, these queries may include agents and other different entity types. An example would be: "Locate the shower jet that Nami loves, with a mirror to its right."

**ScanNet**  We employed ScanNet's validation set (312 scenes) for evaluation. Since ScanNet only includes objects, we emulated agents, induced their abstract relationships to objects, captured spatial relationships between objects, and extracted object features via OVIR-3D before integrating the dataset into our evaluation pipeline. Resource limitations prevented manual labeling of scenes; hence, we synthetically generated 62000 queries (approx.) for evaluation (details in Appendix E.1).

**DOVE-G**  We created DOVE-G to support open-vocabulary queries within scenes using natural language. Each scene includes manually labeled ground truth and 50 original natural language queries ($L_q$). Using LLMs, we expanded this by generating four extra sets of queries, totaling 250 queries per scene and 4000 overall to test OVSG's capabilities with diverse language expressions.

**ICL-NUIM**  To thoroughly compare our method, notably with ConceptFusion, we utilized the ICL-NUIM dataset[8]. We have created 359 natural language queries for the 'Whole Query' category and 190 natural language queries for the 'Object-only Query'. It should be noted that our approach is not merely a superficial addition of another dataset; instead, we have adapted and generated natural language queries for each scene within ICL-NUIM, emulating our methodology with DOVE-G. To adapt it to our framework, we performed similar preprocessing steps as with DOVE-G, importantly manually labeled ground-truth annotations and leveraging OVIR-3D for feature extraction. Using this dataset, we demonstrate the superiority of our proposed method over ConceptFusion, especially concerning complex natural language queries that hinge on multiple relationships as context.

**Evaluation Metrics**

For each query, we evaluated the system's performance using three distinct metrics:

- **IoU$_{BB}$** For each query, this measures the 3D bounding box IoU between the ground truth and the top-k candidates yielded by our system.

- **IoU$_{3D}$** For each query, this measures the IoU between the point cloud indices of the ground truth instance and the predicted instance.

- **Grounding Success Rate** For each scene, this measures the fraction of queries where the system's predictions accurately match the ground truth given that the overlap is significant(**IoU$_{BB}$** $\geq 0.5$ or **IoU$_{3D}$** $> 0.5$). The overlap threshold can be adjusted to alter the strictness of the success criteria.

We reported the Top1 and Top3 Grounding Success Rates and average IoU scores for each scene, reflecting the performance of our system in the Top-k results returned for each query.

| Query Type | # Queries | Metric | Avg. Top1 Scores per Query | | | | Avg. Top3 scores per Query | | | |
|---|---|---|---|---|---|---|---|---|---|---|
| | | | OVIR-3D | OVSG-J | OVSG-S | OVSG-L (Ours) | OVIR-3D | OVSG-J | OVSG-S | OVSG-L (Ours) |
| Object-only | 18,683 | IoU$_{BB}$ | 0.38 | 0.15 | 0.4 | **0.51** | 0.52 | 0.4 | 0.52 | **0.55** |
| | | IoU$_{3D}$ | 0.38 | 0.22 | 0.44 | **0.55** | - | - | - | - |
| | | Grounding Success Rate$_{BB}$ | 38.52 | 15.29 | 40.99 | **52.18** | 52.95 | 41.25 | 53.6 | **56.25** |
| | | Grounding Success Rate$_{3D}$ | 45.13 | 17.22 | 47.79 | **60.35** | - | - | - | - |
| Whole | 20,173 | IoU$_{BB}$ | 0.37 | 0.22 | 0.44 | **0.55** | 0.53 | 0.45 | 0.55 | **0.57** |
| | | IoU$_{3D}$ | 0.39 | 0.16 | 0.41 | **0.53** | - | - | - | - |
| | | Grounding Success Rate$_{BB}$ | 38.56 | 24.33 | 47.77 | **58.85** | 56.28 | 47.84 | 59.87 | **61.6** |
| | | Grounding Success Rate$_{3D}$ | 43.86 | 24.83 | 51.22 | **63.88** | - | - | - | - |

Table 1: Performance of OVSG on ScanNet

**Baselines**  We assessed five methods in our study. The SOTA open-vocabulary grounding method, OVIR-3D, is our primary baseline as it will not leverage any inter-notion relations, providing a comparative measure for the effectiveness of contextual information integration in the other methods. Unlike OVIR-3D, ConceptFusion integrates spatial relationships implicitly. The other three methods, namely OVSG-J, OVSG-S, and OVSG-L (for Jaccard coefficient, Szymkiewicz-Simpson index, and Likelihood, respectively) implement Context-Aware Entity Grounding using different sub-graph matching techniques, as detailed in Section 3.4.

| Query Type | # Queries | Metric | Avg. Top1 Scores per Query | | | | Avg. Top3 scores per Query | | | |
|---|---|---|---|---|---|---|---|---|---|---|
| | | | OVIR-3D | OVSG-J | OVSG-S | OVSG-L (Ours) | OVIR-3D | OVSG-J | OVSG-S | OVSG-L (Ours) |
| Object-only | 320 | $IoU_{BB}$ | 0.37 | 0.14 | 0.39 | **0.49** | 0.57 | 0.36 | **0.56** | 0.56 |
| | | $IoU_{3D}$ | 0.41 | 0.14 | 0.43 | **0.54** | - | - | - | - |
| | | Grounding Success Rate$_{BB}$ | 36.56 | 13.75 | 39.06 | **48.44** | 58.12 | 34.06 | 56.25 | **56.56** |
| | | Grounding Success Rate$_{3D}$ | 49.69 | 18.44 | 53.13 | **67.82** | - | - | - | - |
| Whole | 400 | $IoU_{BB}$ | 0.35 | 0.2 | 0.41 | **0.51** | 0.55 | 0.41 | 0.55 | **0.56** |
| | | $IoU_{3D}$ | 0.37 | 0.21 | 0.43 | **0.55** | - | - | - | - |
| | | Grounding Success Rate$_{BB}$ | 35.5 | 23.0 | 44.75 | **54.25** | 56.0 | 41.0 | 56.75 | **57.0** |
| | | Grounding Success Rate$_{3D}$ | 41.5 | 25.25 | 50.25 | **65.75** | - | - | - | - |

Table 2: Performance of OVSG on DOVE-G

| Query Type | # Queries | Metric | Avg. Top1 Scores per Query | | | | | | Avg. Top3 scores per Query | | | |
|---|---|---|---|---|---|---|---|---|---|---|---|---|
| | | | ConceptFusion (w/o rel) | ConceptFusion | OVIR-3D | OVSG-J | OVSG-S | OVSG-L (Ours) | OVIR-3D | OVSG-J | OVSG-S | OVSG-L (Ours) |
| Object-only | 190 | $IoU_{BB}$ | - | - | 0.32 | 0.18 | 0.37 | **0.5** | 0.55 | 0.49 | 0.55 | **0.56** |
| | | $IoU_{3D}$ | 0.13 (0.3) | 0.06 (0.15) | 0.37 | 0.19 | 0.41 | **0.56** | - | - | - | - |
| | | Grounding Success Rate$_{BB}$ | - | - | 35.26 | 16.84 | 38.95 | **51.6** | 54.74 | 48.95 | 54.74 | **55.79** |
| | | Grounding Success Rate$_{3D}$ | 7.37 (41.18) | 2.64 (14.71) | 48.95 | 22.64 | 51.58 | **68.95** | - | - | - | - |
| Whole | 359 | $IoU_{BB}$ | - | - | 0.33 | 0.34 | 0.47 | **0.61** | 0.62 | 0.56 | **0.64** | 0.64 |
| | | $IoU_{3D}$ | - | - | 0.35 | 0.035 | 0.49 | **0.64** | - | - | - | - |
| | | Grounding Success Rate$_{BB}$ | - | - | 39.28 | 44.29 | 59.61 | **74.09** | 72.42 | 66.3 | 74.09 | **74.65** |
| | | Grounding Success Rate$_{3D}$ | - | - | 45.97 | 44.29 | 61.84 | **78.56** | - | - | - | - |

Table 3: Performance of OVSG & ConceptFusion on ICL-NUIM

## 4.2 Performance

**ScanNet** Table 1 averages results across 312 ScanNet scenes. Contextual data greatly improved entity grounding, with graph similarity variants (OVSG-S, OVSG-L) surpassing OVIR-3D, especially in scenes with repetitive entities like bookstores. More details are in Appendix E.

**DOVE-G** Table 2 averages performance over DOVE-G scenes for five query sets. OVSG-L consistently led, further detailed in Appendix F.3. While OVSG-J and OVSG-S were competitive in some scenes, OVSG-L was generally superior. OVIR-3D shined in the Top3 category, especially since DOVE-G scenes had fewer repetitive entities. Additional insights in Appendix F.

**ICL-NUIM** Table 3 shows ICL-NUIM results with OVSG-L outperforming other methods, especially in the 'Whole Query' segment, contrasting with ScanNet and DOVE-G performances. ConceptFusion's performance was inconsistent across ICL-NUIM scenes (see Appendix G.3), with notable success in one scene (highlighted in orange in Table 3). Simplified queries improved ConceptFusion's results, as depicted in the 'ConceptFusion (w/o rel)' column. Due to its point-level fusion approach, we evaluated different point thresholds and found optimal results at the Top 1500 points. Metrics like $IoU_{BB}$ are not applicable for ConceptFusion. Further details on ICL-NUIM are in Appendix G. Despite ConceptFusion's strategy to avoid motion-blurred ScanNet scenes [32], its efficacy was still suboptimal in certain clear scenes.

Apart from these results, we also provide vocabulary analysis on OVSG as well as two robot experiments. Due to space limits, we put them to Appendices C and D.

## 5 Conclusion & Limitation

Although we have demonstrated the effectiveness of the proposed OVSG in a set of experiments, there still remains three major limitations for our current implementation. First, OVSG heavily relies on an open-vocabulary fusion system like OVIR-3D, which may lead to missed queries if the system fails to identify an instance. Second, the current language processing system's strong dependence on LLMs exposes it to inaccuracies, as any failure in parsing the query language may yield incorrect output. Third, as discussed in Section 3.4, calculating graph likelihood by multiplying nodes and edges likelihoods may not be optimal, as likelihoods from distinct types might carry varying levels of importance and distribution. Accurately balancing these factors remains a challenge for future research, as our efforts with a GNN have not yielded satisfactory results.

Despite the aforementioned areas for improvement, we observe that OVSG significantly improves context-aware entity grounding compared to existing open-vocabulary semantic methods. Since OVSG only requires natural language as the query input, we believe it holds great potential for seamless integration into numerous existing robotics systems.

**Acknowledgments**

This work is supported by NSF awards 1846043 and 2132972.

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

# A Prompt Engineering for Query Parse

As Chain of Thoughts (COT) has demonstrated, by providing with a series of detailed examples, we can guide large language model to generate our desired output while maintaining some format requirement. The design of these examples are also known as prompt engineering.

## A.1 Prompt Example Illustration

Consider this natural language query as an example: "Could you point out Zoro's go-to cup, which we usually keep to the right of our espresso machine, on the left of the trash can, and in front of the coffee kettle?"

In this query, the user is asking about the location of a cup, which has three different spatial relationships with other reference entities and one abstract relationship with a user named Zoro.

The desired output we provided is shown as below:

> There are three notions here: zoro, cup, espresso machine, trash can, coffee kettle. I can only use the relation provided.
> The query target is cup. The relationship between zoro and cup is like. This relationship is a abstract relationship.
> The relationship between cup and espresso machine is right to. This relationship is a spatial relationship.
> The relationship between cup and trash can is left to. This relationship is a spatial relationship.
> The relationship between coffee kettle and cup is behind. This relationship is a spatial relationship.
> The notion, target, and relationship are:
> "'
> target @ cup {object}
> zoro {user} – like [abstract] – cup {object}
> cup {object} – right to [spatial] – espresso machine {object}
> cup {object} – left to [spatial] – trash can {object}
> coffee kettle {object} – behind [spatial] – cup {object}
> '''

This example starts from some reasoning process in natural language, and ends with a structured output which can be parsed by code. A breakdown of the structure is as follows:

**target @ cup {object}**: This line specifies the target object, which is a cup.

**zoro {user} – like [abstract] – cup {object}**: This line represents a relationship between a user named Zoro (user) and the cup (an object) that Zoro likes the cup (Zoro's favorite). In our current implementation, like is a relation of type *abstract*.

**cup {object} – right to [spatial] – espresso machine {object}**: This line represents a spatial relationship between the cup (an object) and the espresso machine (an object). The cup is positioned to the right of the espresso machine.

**cup {object} – left to [spatial] – trash can {object}**: This line represents a spatial relationship between the cup (an object) and the trash can (an object). The cup is positioned to the left of the trash can.

**coffee kettle {object} – behind [spatial] – cup {object}**: This line describes a spatial relationship between the coffee kettle (an object) and the cup (an object). The coffee kettle is positioned behind the cup.

## A.2   More prompt examples

Before asking the LLM to process the real user input, we will first input around 10 examples as a prompt to control the output format. We select a few examples to show here.

> Question: I want to get the cracker box around the table in the kitchen.
> There are three notions here: cracker box, table, and kitchen. I can only use the relation provided.
> The query target is the cracker box.
> This is a query for an object of the known category: cracker box.
> The relationship between the cracker box and the table is 'near'. This relationship is a spatial relationship.
> The relationship between the table and the kitchen is 'in'. This relationship is a spatial relationship.
> The notion, target, and relationship are:
> '''
> target @ cracker box object
> cracker box object – near [spatial] – table object
> table object – in [spatial] – kitchen region
> '''

> Question: Bring Tom his favorite drink.
> There are two notions here: Tom and drink. I can only use the relation provided.
> This is a query for an object of a known category: drink.
> The relationship between me and drink is 'like'. This relationship is a spatial relationship.
> The query target is 'drink'.
> The notion, target, and relationship are:
> '''
> target @ drink object
> Tom user – like [spatial] – drink object
> '''

> Question: Can you find Marry's favourite coffee cup? It might be at the kitchen.
> There are three notions here: Mary, coffee cup, and kitchen.
> This is a query for object of known category: coffee cup.
> The relationship between Mary and coffee cup is like. This relationship is a user relationship.
> The relationship between coffee cup and kitchen is in. This relationship is a spatial relationship.
> The query target is coffee cup.
> The notion, target, and relationship are:
> '''
> target @ coffee cup object
> Mary user – like [user] – coffee cup object
> coffee cup object – in [spatial] – kitchen region
> '''

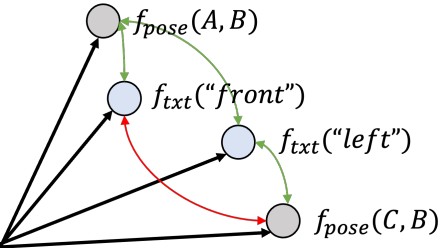

Figure 3: The illustration highlights the non-linear nature of the spatial language feature. Let's assume both the spatial pose feature and spatial text feature can be represented within a singular linear space. For instance, consider A being to the left and in front of B, while C is to the left but behind B. The pose feature for A relative to B should align closely with the text features "left" and "front". Conversely, the pose feature for C relative to B should be close to the text feature "left" but distant from "front". If all these features were mapped onto a linear space, the pose feature $f_{pose}(A, B)$ would paradoxically be both near and far from $f_{pose}(C, B)$.

# B  Spatial Relationship Prediction Pipeline

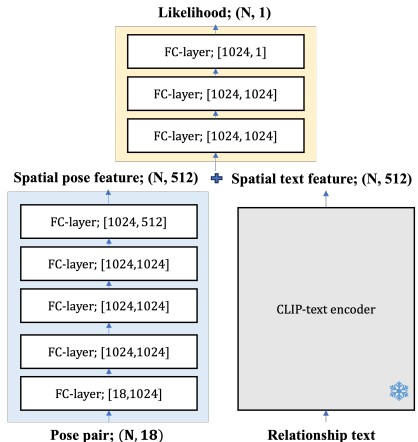

Figure 2: Architecture of spatial-language encoder and predictor. The blue block is the spatial pose encoder, and the yellow block is the spatial relationship predictor.

The Spatial Relationship Predictor module aims to estimate the likelihood between pose pairs and language descriptions. Given that there is no standard solution to this spatial-language alignment challenge, we have developed our own encoder-predictor structure.

**Network Structure** The input for the spatial pose encoder (depicted as a blue block in Figure 2) is a pose pair defined by (N, 18). An entity's pose in the OVSG is characterized by the boundaries and center of its bounding box, specifically $(x_{min}, y_{min}, z_{min}, x_{max}, y_{max}, z_{max}, x_{center}, y_{center}, z_{center})$. We employ a five-layer MLP to encode this pose pair into a spatial pose feature. For the encoding of the spatial relationship description, we utilize the CLIP-text encoder, converting it into a 512-dimensional vector.

**Distance Design** These encoders serve as the foundation for constructing the OVSG. When performing sub-graph matching, the predictor head estimates the distance between the spatial pose feature and the spatial text feature. We do not use cosine distance because the spatial relationship is highly non-linear. Figure 3 illustrates why cosine distance is not sufficiently discriminative for spatial-language alignment.

**Training process** We train this encoder and predictor module using supervised learning. The training data is generated synthetically. We manually defined 8 different single spatial relationships, i.e. left, right, in front of, behind, in, on, above, under. From these 8 basic spatial relationships, we can generated more than 20 different meaningful combinations, e.g. "on the right side", "at the left front part". Each combinations can also have more than one descriptions. Finally, we collected 90 descriptions in total. The training loss we used is a binary cross entropy loss.

## C  Robot application

**Manipulation**  In order to exemplify the utility of OVSG in real-world manipulation scenarios, we devised a complex pick-and-place experiment. In this task, the robot is instructed to select one building block and position it on another. The complexity of the task stems from the multitude of blocks that are identical in both shape and color, necessitating the use of spatial context for differentiation. Each task consists of a picking action and a placing action. We formulated nine distinct tasks for this purpose (please refer to Appendix C.1 for detailed setup). The effectiveness of the manipulation task was evaluated by comparing the success rate achieved by OVIR-3D and our newly proposed OVSG-L. The outcome of this comparative study is depicted in the accompanying table. The results demonstrate that our innovative OVSG-L model significantly enhances the object grounding accuracy in manipulation tasks involving a high prevalence of identical objects. This improvement highlights the potential of OVSG-L in complex manipulation scenarios, paving the way for further exploration in the field of robotics.

| object | shoe | bottle | chair | trash can#1 | trash can#2 | drawer | cloth |
|---|---|---|---|---|---|---|---|
| success rate (%) | 100.0 | 100.0 | 100.0 | 100.0 | 100.0 | 100.0 | 0.0 |

Table 4: Success rate of object navigation task

| Method | scene1 | scene2 | scene3 |
|---|---|---|---|
| OVIR-3D (%) | 0.0 | 0.75 | 0.33 |
| OVSG (%) | 0.88 | 0.75 | 0.75 |

Table 5: Success rate of manipulation task

**Navigation**  We conducted a system test on a ROSMASTER R2 Ackermann Steering Robot for an object navigation task. The detailed setup can be found in Appendix C.2. We provided queries for seven different objects within a lab scene, using three slightly different languages to specify each object. These queries were then inputted into OVSG, and the grounded positions of the entities were returned to the robot. We considered the task successful if the robot's final position was within 1 meter of the queried objects. The results are presented in Table 4. From the table, it is evident that the proposed method successfully located the majority of user queries. However, there was one query that was not successfully located: "The cloth on a chair in the office." In this case, we found that OVIR-3D incorrectly recognized the cloth as part of a chair, resulting in the failure to locate it.

### C.1  Manipulation Experiment Setup

**Robot Setup**  All evaluations were conducted using a Kuka IIWA 14 robot arm equipped with a Robotiq 3-finger adaptive gripper. The arm was augmented with an Intel Realsense D435 camera, which was utilized to capture the depth and color information of the scene in an RGB-D format, offering a resolution of 1280 x 720. The gripper operated in "Pinch Mode," whereby the two fingers on the same side of the gripper bent inward.

To initiate the process, the robot arm was employed to position the camera above the table, orienting it in a downward direction. Subsequently, the RGB-D data, along with a query specifying the object to be picked and a target object for placement, were inputted into the OVSG system. Upon acquiring the bounding box of the query object, the robot gripper was directed to move towards the center coordinates of the target box by utilizing the ROS interface of the robot arm.

**Block building task**  To evaluate the application of the proposed method in real-world manipulation tasks, we designed a block-building task. The task is to pick one building block from a set of building blocks and place it on another building block. The picking block and placing block are separately specified by a different natural language query. The difficulty of this task is that each building block has many repeats around it so we have to use spatial context to specify the building block. And we need to succeed twice in a row to complete a task.

| Object | Query Id | Query |
|---|---|---|
| shoe | 1 | A shoe that is in front of the monitor and in the office. |
| | 2 | A shoe positioned both before the monitor and within the office. |
| | 3 | A shoe rests in front of the monitor. The shoe is inside the office. |
| bottle | 4 | The bottle that is right to Tom's keyboard |
| | 5 | The bottle to the right of Tom's keyboard. |
| | 6 | The bottle positioned to the right of a keyboard which belongs to Tom. |
| chair#1 | 7 | The chair that is behind the TV. |
| | 8 | The chair is situated behind the TV |
| | 9 | The chair with a TV in front of it. |
| chair#2 | 10 | The chair that is in front of the car. |
| | 11 | The chair rests before a car. |
| | 12 | The chair with a car behind it. |
| trash can#1 | 13 | The trash can that is behind the refrigerator. |
| | 14 | The trash can which can be found behind the refrigerator. |
| | 15 | The trash can that is situated at the rear of the refrigerator. |
| trash can#2 | 16 | The trash can that is in the lab and under a table. |
| | 17 | The trash can is located in the lab and beneath a table. |
| | 18 | The trash can is situated within the lab, positioned under a table. |
| drawer | 19 | The drawer that is behind a box in the office. |
| | 20 | The drawer is positioned behind a box in the office. |
| | 21 | The drawer is situated at the rear of a box within the office. |
| cloth | 22 | The cloth that is on a chair |
| | 23 | The cloth is resting on a chair |
| | 24 | The cloth is positioned atop a chair. |

Table 6: Queries for navigation task

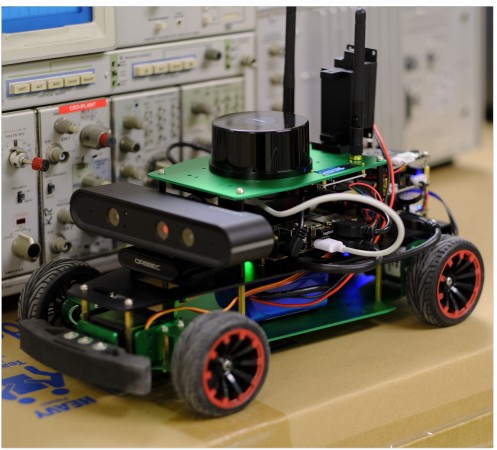 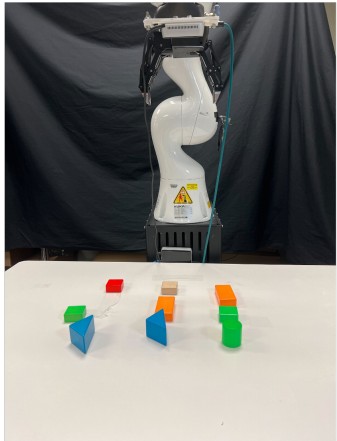

Figure 4: The left one is the robot for our navigation task, ROSMASTER R2 Ackermann Steering Robot. The right one is the robot for our manipulation task, KUKA IIWA 14

## C.2 Navigation Experiment Setup

**Robot Setup** All evaluations were conducted using a ROSMASTER R2 Ackermann Steering Robot. For perception, we utilized an Astra Pro Plus Depth Camera and a YDLidar TG 2D lidar sensor, both mounted directly onto the robot. The robot is equipped with a built-in Inertial Measurement Unit (IMU) and wheel encoder. The Astra camera provides a video stream at a resolution of 720p at 30 frames per second, and the lidar operates with a sampling frequency of 2000 Hz and a scanning radius of approximately 30 meters. The overall configuration of the setup is depicted in Figure 4.

**Demonstrations and Execution** Prior to the evaluation process, we employed an Intel RealSense D455 camera and ORB-SLAM3 [33] to generate a comprehensive map of the environment. This generated both the RGB-D and pose data, which could be subsequently fed into the Open-vocabulary

pipeline. For the demonstration of locating with the Open-Vocabulary 3D Scene Graph (OVSG), we developed a 3D to 2D conversion tool. This tool takes the point cloud from the comprehensive 3D map and converts it into a 2D map by selecting a layer of points at the height of the lidar. The resultant 2D map could then be utilized by the ROSMASTER R2 Ackermann Steering robot for navigation. To achieve goal-oriented navigation, we incorporated the Robot Operating System (ROS) Navigation stack and integrated it with the Timed Elastic Band (TEB) planner. The initial step involved establishing a pose within the environment. Subsequently, the Adaptive Monte Carlo Localization (AMCL) leveraged lidar scan inputs and IMU data to provide a robust estimate of the robot's pose within the map. The move base node, a key component of the ROS navigation stack, used the converted map and the item's position provided by the OVSG and conversion tool to formulate a comprehensive global plan targeting the goal position. Concurrently, the TEB local planner consolidated information about ROSMASTER R2's kinematics and lidar input to generate a short-term trajectory. The result was a locally optimized, time-efficient plan that adhered to the robot's pre-set velocity and acceleration limits. The plan also included obstacle avoidance capabilities, enabling the robot to identify and circumvent barriers detected by the lidar system.

**Object navigation task** To evaluate the application of OVSG in real-world navigation problems, a language-based object navigation task is proposed. We selected seven different objects inside a laboratory. Each object is paired with three different queries. All queries for three objects are listed in Table 6.

| Vocab Set | Top1 Grounding Success Rate$_{BB}$ | | | | Top3 Grounding Success Rate$_{BB}$ | | | |
|---|---|---|---|---|---|---|---|---|
| | OVIR-3D | OVSG-J | OVSG-S | OVSG-L (Ours) | OVIR-3D | OVSG-J | OVSG-S | OVSG-L (Ours) |
| #1 | 35.14 | 23.71 | 42.29 | **47.14** | 53.14 | 40.86 | 55.71 | **57.14** |
| #2 | 35.71 | 27.43 | 45.43 | **51.71** | 58.86 | 46.86 | 62.57 | **62.57** |
| #3 | 32.29 | 26.00 | 43.43 | **48.86** | 58.86 | 47.71 | 60.57 | **62.00** |
| #4 | 33.14 | 20.29 | 42.29 | **44.86** | 55.71 | 42.86 | 57.14 | **56.57** |
| #5 | 42.86 | 29.71 | 52.57 | **57.43** | 62.57 | 52.29 | 65.71 | **65.43** |
| Overall | 35.83 | 25.43 | 45.20 | **50.00** | 57.83 | 46.12 | 60.34 | **60.74** |
| | Top1 IoU$_{BB}$ | | | | Top3 IoU$_{BB}$ | | | |
| | OVIR-3D | OVSG-J | OVSG-S | OVSG-L (Ours) | OVIR-3D | OVSG-J | OVSG-S | OVSG-L (Ours) |
| #1 | 0.31 | 0.18 | 0.36 | **0.42** | 0.50 | 0.39 | 0.51 | **0.53** |
| #2 | 0.31 | 0.22 | 0.38 | **0.44** | 0.53 | 0.43 | 0.55 | **0.56** |
| #3 | 0.29 | 0.21 | 0.37 | **0.43** | 0.52 | 0.43 | 0.54 | **0.56** |
| #4 | 0.30 | 0.17 | 0.36 | **0.40** | 0.51 | 0.39 | 0.52 | **0.52** |
| #5 | 0.38 | 0.24 | 0.44 | **0.50** | 0.57 | 0.47 | 0.59 | **0.60** |
| Overall | 0.32 | 0.20 | 0.38 | **0.44** | 0.53 | 0.42 | 0.54 | **0.56** |

Table 7: Performance comparison against five different varied open-vocabulary sets

| Relationship Set | Top1 Grounding Success Rate$_{BB}$ | | | | Top3 Grounding Success Rate$_{BB}$ | | | |
|---|---|---|---|---|---|---|---|---|
| | OVIR-3D | OVSG-J | OVSG-S | OVSG-L (Ours) | OVIR-3D | OVSG-J | OVSG-S | OVSG-L (Ours) |
| #1 | 35.14 | 19.50 | 36.00 | **41.38** | 54.12 | 40.88 | 53.88 | **58.12** |
| #2 | 35.71 | 21.50 | 36.75 | **44.62** | 59.12 | 47.62 | 62.38 | **62.88** |
| #3 | 32.29 | 23.25 | 38.25 | **42.12** | 58.88 | 49.88 | 61.12 | **59.88** |
| #4 | 33.14 | 18.75 | 36.50 | **40.38** | 56.38 | 43.12 | 55.38 | **56.38** |
| #5 | 42.86 | 22.75 | 43.25 | **47.88** | 62.38 | 52.38 | 65.12 | **65.38** |
| Overall | 35.83 | 21.15 | 38.15 | **43.28** | 58.18 | 46.76 | 59.58 | **60.53** |
| | Top1 IoU$_{BB}$ | | | | Top3 IoU$_{BB}$ | | | |
| | OVIR-3D | OVSG-J | OVSG-S | OVSG-L (Ours) | OVIR-3D | OVSG-J | OVSG-S | OVSG-L (Ours) |
| #1 | 0.30 | 0.17 | 0.31 | **0.36** | 0.50 | 0.38 | 0.48 | **0.53** |
| #2 | 0.30 | 0.19 | 0.32 | **0.39** | 0.54 | 0.43 | 0.55 | **0.56** |
| #3 | 0.29 | 0.20 | 0.33 | **0.38** | 0.53 | 0.45 | 0.54 | **0.54** |
| #4 | 0.30 | 0.16 | 0.30 | **0.36** | 0.52 | 0.40 | 0.50 | **0.52** |
| #5 | 0.38 | 0.22 | 0.37 | **0.42** | 0.57 | 0.48 | 0.58 | **0.60** |
| Overall | 0.32 | 0.18 | 0.33 | **0.38** | 0.53 | 0.43 | 0.53 | **0.55** |

Table 8: Performance comparison against five different varied relationship sets

# D Open-Vocabulary Analysis

Having presented insights on our system's performance on natural language queries for DOVE-G (as shown in Table 2), we proceed to deepen our investigation into the system's resilience across diverse query sets. To accomplish this, we instead average the results from all scenes for each of the five vocabulary sets (refer to Table 7). By doing so, we aim to provide a robust evaluation of our system's performance across a variety of query structures and word choices, simulating the varied ways in which users may interact with our system. In addition to experimenting with object vocabulary variations (a 'coffee maker' to 'espresso machine' or 'coffee brewer'), and altering the order of entity referencing in the query, we also studied the impact of changing relationship vocabulary. In this experimental setup, the LLM is not bound to map relationships to a pre-determined set as before. Instead, the graph-based query contains a variety of relationship vocabulary. To illustrate, consider the queries "A is to the left back corner of B" and "A is behind and left to B". Previously, these relationships would map to a fixed relation like 'left and behind'. Now, 'front and left' as interpreted by the LLM can variate to 'leftward and ahead', 'northwest direction', or 'towards the front and left', offering a broader range of relationship descriptions. The evaluation results for these query sets are presented in Table 8.

**Varying object names** Across all evaluated vocabulary sets, OVSG-L demonstrates the highest Top1 and Top3 Grounding Success Rates$_{BB}$, outperforming the remaining methods. This pattern also persists for scores in the $IoU_{BB}$ category. Notably, OVSG-L's Grounding Success Rates span from 44.86% to 57.43% for Top1, and 56.57% to 65.43% for Top3. All in all, contextual understanding of the target again proves to improve results from 35.83% (OVIR-3D) to 50% (OVSG-L) for Top1 Grounding Success Rate$_{BB}$ and 0.32 to 0.44 for the Top1 $IoU_{BB}$.

**Varying relationships** As shown in Table 8, we observe a noticeable decrease in performance for the methods under the OVSG framework (compared to Table 7). This is likely due to the increased complexity introduced by the varied word choices for edges (relationships) in the sub-graph being matched. Despite this, two of the OVSG methods still outperform the OVIR-3D method, with the OVSG-L method delivering the strongest results.

# E   More on ScanNet

## E.1   Synthetic Query Generation for ScanNet

In the ScanNet dataset, each scene comes with ground-truth labels for its segmented instances or objects. We began by calculating the spatial relationships between these ground-truth objects or entities. Subsequently, agents were instantiated into the scene, and abstract relationships were randomly established between the agents and the entities present in the scene. After generating the OVSG for each scene, our next step involved the creation of graph-based queries (refer to syntax and details in Appendix A) for evaluation purposes. For each of these queries, we randomly selected reference entities from the OVSG that shared a relationship with the target entity. This formed the basis of the synthetic generation of the graph-based queries for the ScanNet dataset.

## E.2   Grounding Success Rate$_{\text{BB}}$

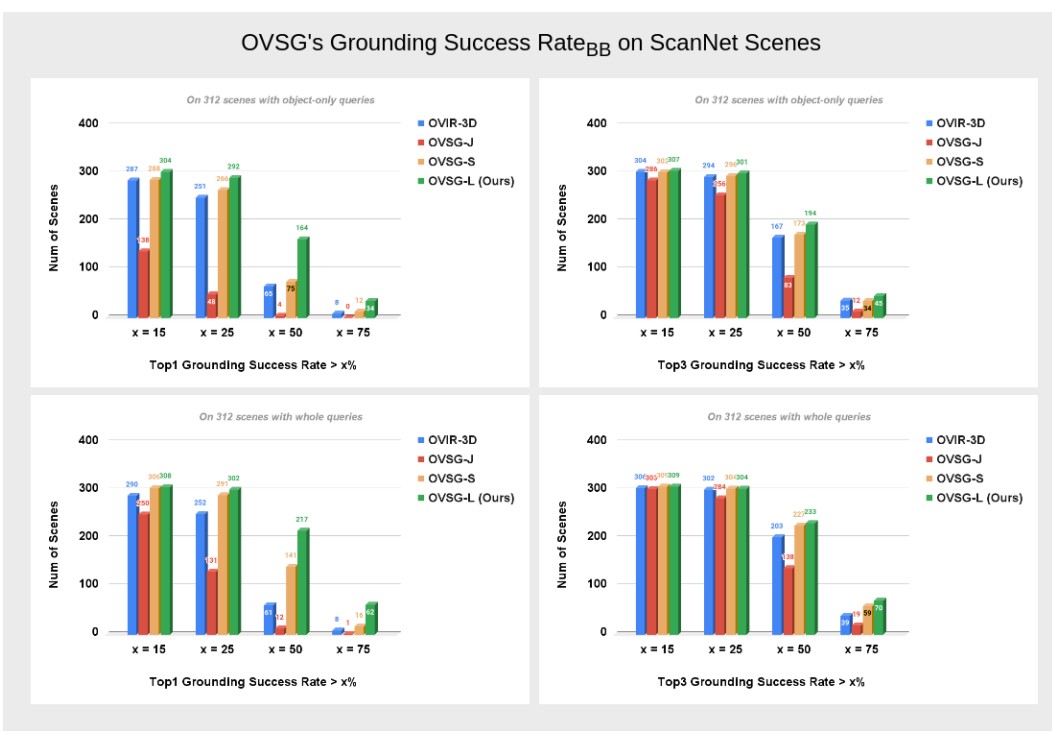

Figure 5: Performance of OVSG w.r.t Grounding Success Rate$_{\text{BB}}$ on ScanNet Scenes

In this section, we provide the number of ScanNet scenes that correspond to various success rate thresholds (at 15%, 25%, 50%, and 75%). We provide four-fold results containing Top1 and Top3 scores for 'Object-only' and 'Whole Query' categories (as shown in Figure 5).

## E.3   Grounding Success Rate$_{\text{3D}}$

In this section, we provide the various success rates for different $\text{IoU}_{\text{3D}}$ thresholds (at 0.15, 0.25, 0.5, and 0.75). We provide two-fold results containing scores for 'Object-only' and 'Whole Query' categories (as shown in Figure 6).

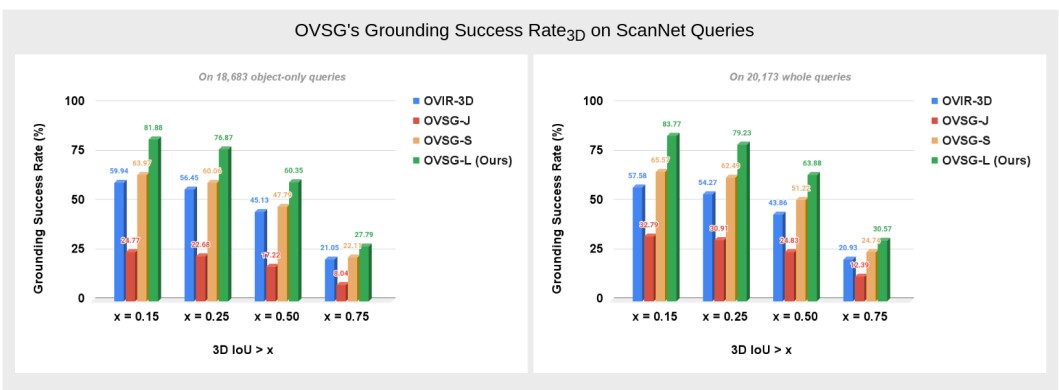

Figure 6: Performance of OVSG w.r.t Grounding Success Rate$_{3D}$ on ScanNet Queries

# F    More on DOVE-G

## F.1    Grounding Success Rate$_{BB}$

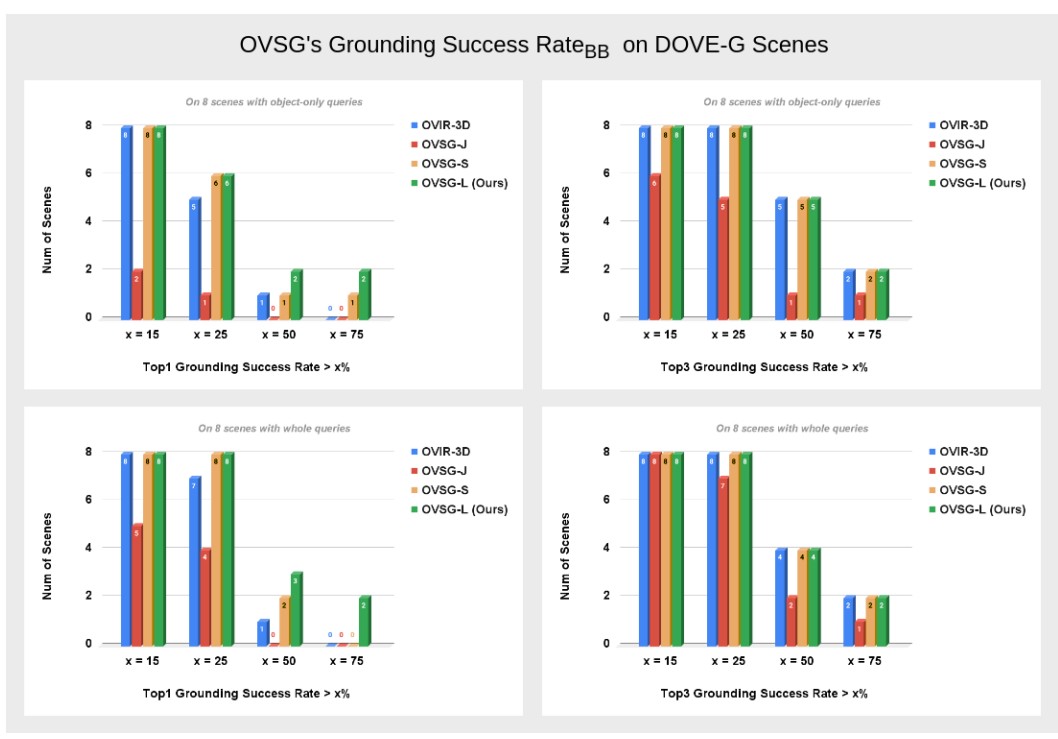

Figure 7: Performance of OVSG w.r.t Grounding Success Rate$_{BB}$ on DOVE-G Scenes

In this section, we provide the number of DOVE-G scenes that correspond to various success rate thresholds (at 15%, 25%, 50%, and 75%). We provide four-fold results containing Top1 and Top3 scores for 'Object-only' and 'Whole Query' categories (as shown in Figure 7).

## F.2    Grounding Success Rate$_{3D}$

In this section, we provide the various success rates for different $\mathbf{IoU_{3D}}$ thresholds (at 0.15, 0.25, 0.5, and 0.75). We provide two-fold results containing scores for 'Object-only' and 'Whole Query' categories (as shown in Figure 8).

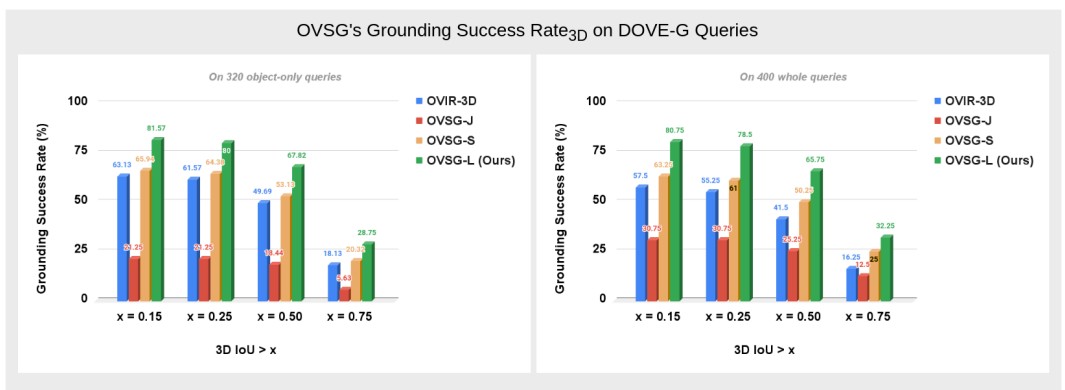

Figure 8: Performance of OVSG w.r.t Grounding Success Rate$_{3D}$ on DOVE-G Queries

## F.3 Performance of the OVSG Framework on Various Scenes in DOVE-G

| Scene | Top1 Grounding Success Rate$_{BB}$ | | | | Top3 Grounding Success Rate$_{BB}$ | | | |
|---|---|---|---|---|---|---|---|---|
| | OVIR-3D | OVSG-J | OVSG-S | OVSG-L (Ours) | OVIR-3D | OVSG-J | OVSG-S | OVSG-L (Ours) |
| Room #1 | 44.4 | 36.8 | 61.2 | **79.6** | 88.4 | 72.8 | 98.4 | **99.6** |
| Kitchenette | 31.0 | 24.0 | 36.0 | **49.0** | 65.0 | 51.0 | 71.0 | **71.0** |
| Bathroom | 44.8 | 23.2 | 57.2 | **58.0** | 59.6 | 41.2 | 66.0 | **69.6** |
| Kitchen | 44.8 | 36.0 | 51.2 | **55.6** | 53.6 | 50.8 | 55.2 | **56.8** |
| Room #2 | 18.0 | 26.0 | 40.0 | **41.6** | 62.0 | 48.0 | **65.2** | 62.8 |
| Computer Lab | 38.0 | 30.8 | 38.0 | **41.6** | **54** | 41.2 | 50.4 | 49.2 |
| Room #3 | 32.8 | 16.8 | 40.4 | **40.4** | 47.2 | 34.4 | **47.2** | 47.2 |
| Hallway | 28.0 | 8.4 | 28.4 | **33.2** | 40.0 | 34.4 | 40.0 | **40.0** |
| Overall | 35.2 | 25.2 | 45.4 | **51.1** | 63.6 | 47.1 | 64.8 | **65.7** |
| | Top1 IoU$_{BB}$ | | | | Top3 IoU$_{BB}$ | | | |
| | OVIR-3D | OVSG-J | OVSG-S | OVSG-L (Ours) | OVIR-3D | OVSG-J | OVSG-S | OVSG-L (Ours) |
| Room #1 | 0.26 | 0.18 | 0.36 | **0.47** | 0.55 | 0.39 | 0.60 | **0.61** |
| Kitchenette | 0.28 | 0.23 | 0.34 | **0.42** | 0.59 | 0.47 | 0.62 | **0.62** |
| Bathroom | 0.30 | 0.13 | 0.36 | **0.36** | 0.41 | 0.30 | 0.44 | **0.46** |
| Kitchen | 0.35 | 0.28 | 0.38 | **0.42** | 0.47 | 0.42 | 0.46 | **0.48** |
| Room #2 | 0.15 | 0.21 | 0.31 | **0.34** | 0.51 | 0.41 | **0.53** | 0.52 |
| Computer Lab | 0.32 | 0.22 | 0.33 | **0.39** | **0.49** | 0.39 | 0.45 | 0.46 |
| Room #3 | 0.34 | 0.13 | 0.37 | **0.39** | 0.50 | 0.36 | 0.49 | **0.50** |
| Hallway | 0.27 | 0.10 | 0.24 | **0.31** | 0.37 | 0.34 | 0.38 | **0.41** |
| Overall | 0.28 | 0.18 | 0.34 | **0.39** | 0.49 | 0.39 | 0.49 | **0.50** |

Table 9: Performance of the OVSG framework on natural language scene queries in DOVE-G

In Table 9, we present the performance of our OVSG framework on natural language scene queries in DOVE-G.

## F.4 50 Sample Natural Language Queries for Scenes in DOVE-G

In Table 10, we provide a list of 50 sample queries for scenes in DOVE-G.

## F.5 More on Scenes in DOVE-G

In Figure 9 and Figure 10, we display eight different scenes included in our DOVE-G dataset.

| Query No. | Natural Language Query |
|---|---|
| 1 | Locate the vanity sink, which is positioned to the right side of a door latch. |
| 2 | Identify the hand basin that has a face cleanser situated in front of it. |
| 3 | Is there a hand basin that has both a facial scrub and hand soap placed in front of it? |
| 4 | I'm looking for a wash-hand basin with a facial scrub directly in front of it and a door latch to its right. |
| 5 | Locate the shower jet that Nami loves, with a mirrored surface to its right and a hair cleanser in front of it. |
| 6 | Can you find the shower sprayer with a face cleanser positioned behind it? |
| 7 | Search for the shower sprayer that has a facial scrub behind it and a vanity mirror to its right. |
| 8 | Identify the travel suitcase located to the right of a backpack and ahead of a water bottle. |
| 9 | Look for a travel suitcase Zoro dislikes, it should be to the right of a book and a water bottle. |
| 10 | Can you find a book positioned to the left of a backpack? |
| 11 | Nami's preferred book should be positioned in front of a chair, can you find it? |
| 12 | Can you identify Zoro's liked book that's situated ahead of a water bottle and another book? |
| 13 | Locate a Carry-on Luggage for me, please. It should have both a water bottle and a rucksack in front. |
| 14 | Is there a trolley bag with a water bottle and a backpack up front, and also a desk chair in its rear? |
| 15 | Find an ergonomic chair for me, but it has to have a textbook situated to its left. |
| 16 | Can you find the workbook that Luffy dislikes and is right of a desk chair? |
| 17 | Is there a headrest that has a reference book positioned in front of it? |
| 18 | Where's the cushion with a coursebook and an ergonomic chair up front? |
| 19 | I'm searching for a backpack with a coursebook on its left. |
| 20 | Where's the table fan with a desk chair behind it? |
| 21 | Can you find a table fan with a computer chair and a reference book behind it? |
| 22 | Where's the reading lamp that's to the left of a travel bag? |
| 23 | Can you spot the reading lamp that's to the left of a travel bag and behind a computer chair? |
| 24 | Is there a reading lamp that's behind a reference book? |
| 25 | Can you find a pedestal fan with a desk chair and a coursebook behind it? |
| 26 | Locate the headrest that's disliked by Nami, but Luffy is indifferent to. |
| 27 | How about a cushion that Nami dislikes, Luffy is neutral to, and Zoro takes a liking to? |
| 28 | Where's the reading lamp that's to the left of a knapsack? |
| 29 | Can you spot the reading lamp that's to the left of a knapsack and behind an ergonomic chair? |
| 30 | Is there a desk lamp that's behind a workbook? |
| 31 | Where's the reading lamp that Nami is fond of? |
| 32 | Can you locate a backpack with a table fan to its left? |
| 33 | Where's the knapsack with a tower fan on its left and Luffy behind? |
| 34 | Can you find a travel bag that has a water bottle on its left? |
| 35 | I'm searching for a backpack with a textbook on its left. |
| 36 | Where's the pedestal fan with a computer chair behind it? |
| 37 | Where is the cup that's nestled to the right of the coffee maker, to the left of the coffee kettle, and in front of the poster? |
| 38 | Identify the toy that's to the right of the espresso machine and to the left of the trash can. |
| 39 | Where's the doll with a cup positioned behind it? |
| 40 | Can you show me the water bottle that Luffy loves and has a coffee cup behind it? |
| 41 | Locate the water bottle that's to the left of the checkerboard. |
| 42 | I want to know about the coffee cup that Zoro loves, which is also behind the keyboard that Nami is behind. |
| 43 | Can you identify the chair that the CPU machine is behind? |
| 44 | Locate the chair that Nami likes and is also behind the CPU machine. |
| 45 | Can you identify the teacup that Luffy loves and is behind the CPU machine? |
| 46 | Is there a computer chair that Zoro doesn't prefer? |
| 47 | Identify a coursebook located ahead of a water bottle. |
| 48 | Is there a workbook sandwiched between two water sipper bottles? |
| 49 | Nami's preferred coursebook should be positioned in front of a desk chair, can you find it? |
| 50 | Zoro's liked reading book, is it placed in front of a water beverage bottle? |

Table 10: List of 50 sample queries for scenes in DOVE-G

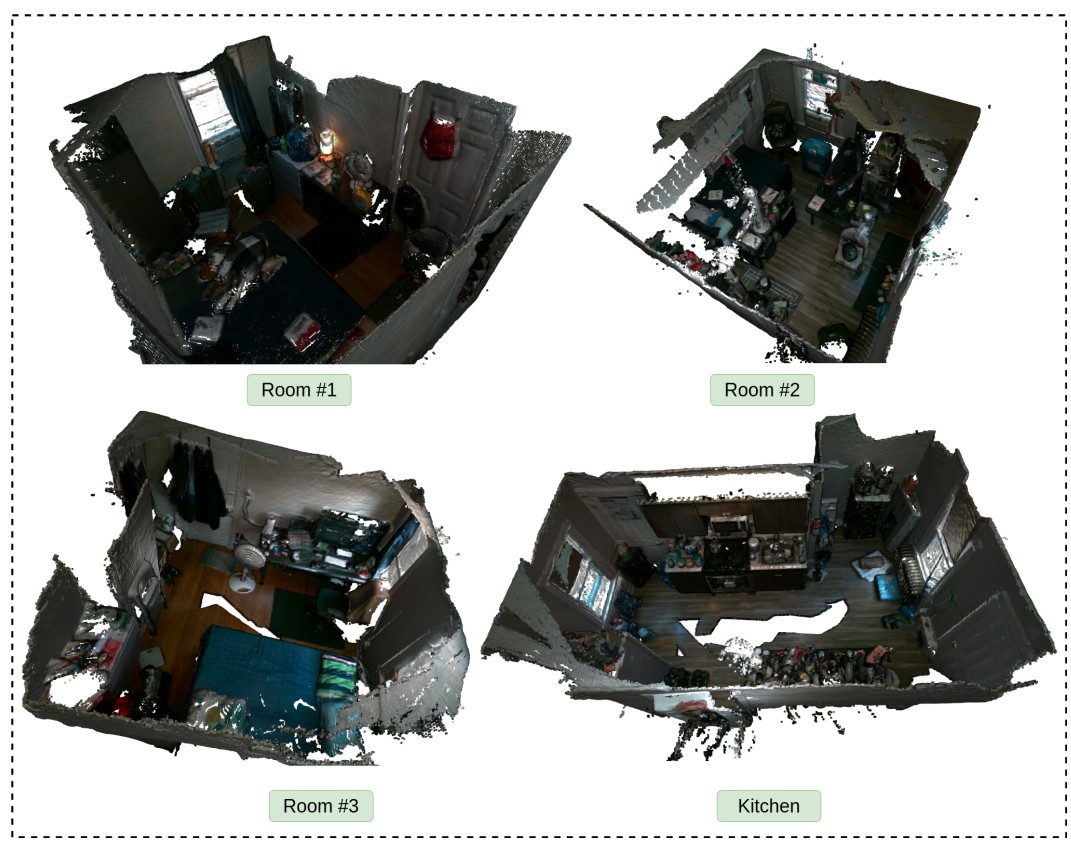

Figure 9: Four of the scenes in DOVE-G

## G   More on ICL-NUIM

### G.1   Grounding Success Rate$_{BB}$

In this section, we provide the number of ICL-NUIM scenes that correspond to various success rate thresholds (at 15%, 25%, 50%, and 75%). We provide four-fold results containing Top1 and Top3 scores for 'Object-only' and 'Whole Query' categories (as shown in Figure 11).

### G.2   Grounding Success Rate$_{3D}$ in comparison to ConceptFusion

In this section, we provide the various success rates for different $IoU_{3D}$ thresholds (at 0.15, 0.25, 0.5, and 0.75). We provide two-fold results containing scores for 'Object-only' and 'Whole Query' categories (as shown in Figure 12).

### G.3   Scene by Scene Grounding Success Rate$_{3D}$ of OVSG & ConceptFusion on ICL-NUIM

Table 11 showcases the 3D Grounding Success Rate of various methods on different scenes in the ICL-NUIM dataset, highlighting the performance metrics across different $IoU_{3D}$ thresholds.

### G.4   Qualitative Performance Comparison between ConceptFusion and OVSG-L

In this section, we are providing qualitative results on sample queries for the methods ConcepFusion and OVSG-L in Figure 13 and Figure 14 respectively.

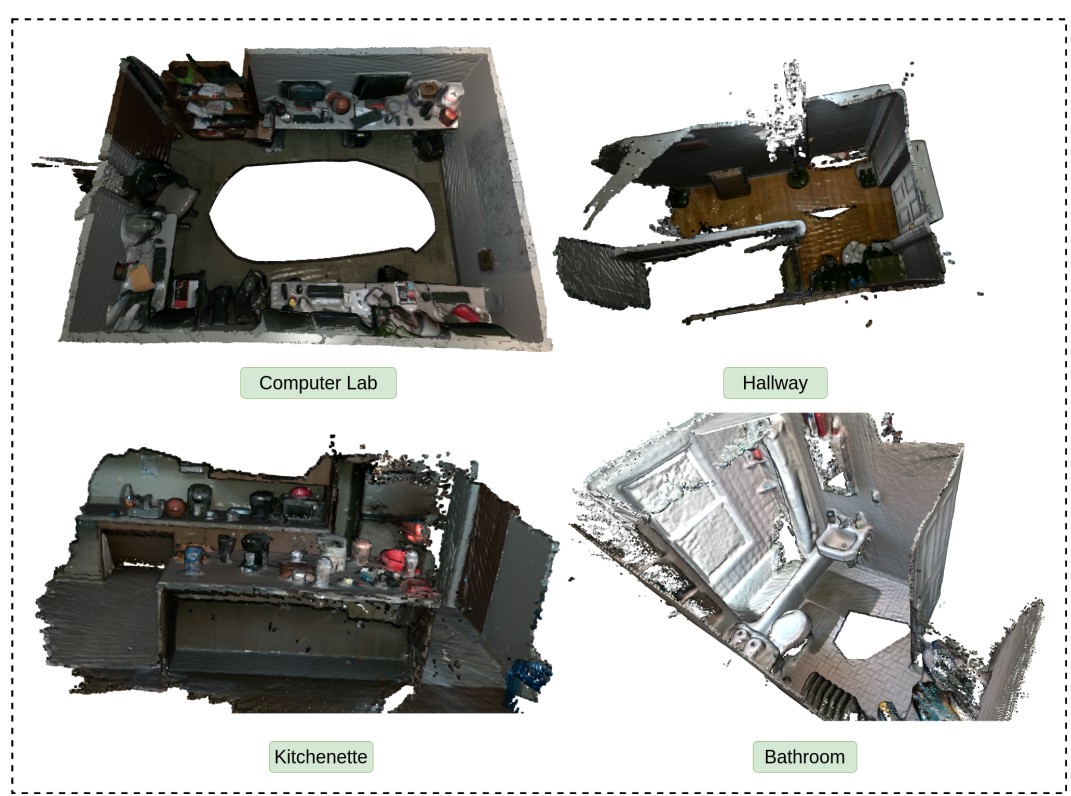

Figure 10: Four of the other scenes in DOVE-G

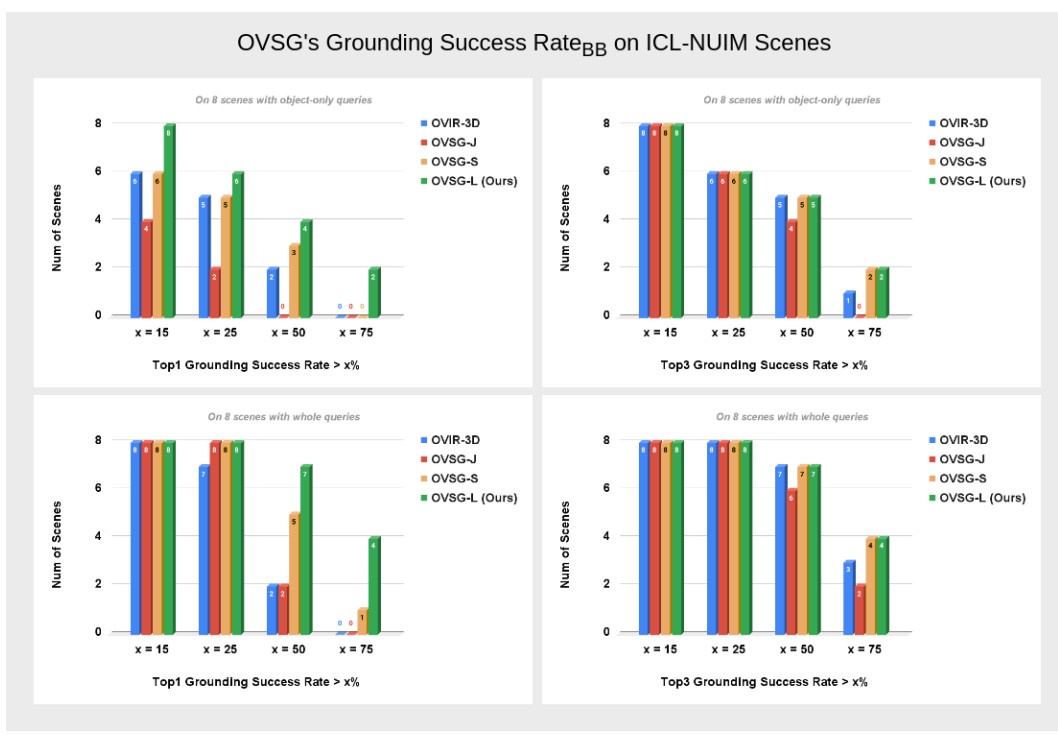

Figure 11: Performance of OVSG w.r.t Grounding Success Rate$_{BB}$ on ICL-NUIM Scenes

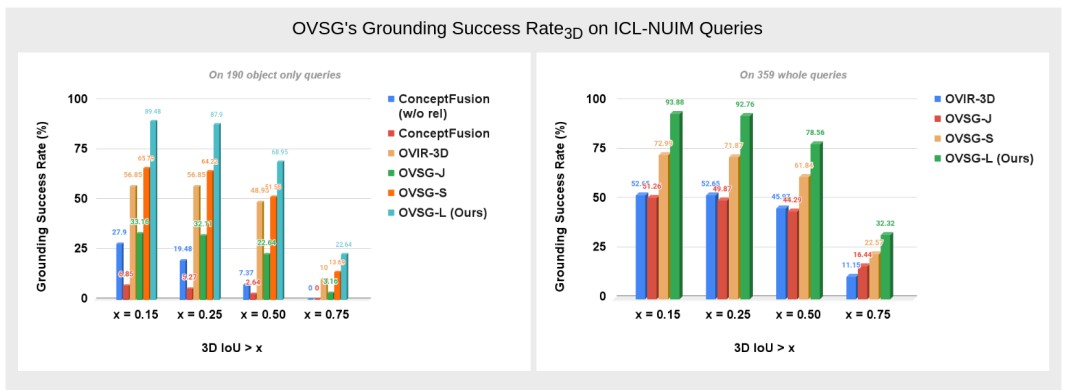

Figure 12: Performance of OVSG & ConceptFusion w.r.t Grounding Success Rate$_{3D}$ on ICL-NUIM Queries

| ICL-NUIM Scene | # Queries | Method | Grounding Success Rate$_{3D}$ | | | |
|---|---|---|---|---|---|---|
| | | | $\mathbf{IoU_{3D} > 0.15}$ | $\mathbf{IoU_{3D} > 0.25}$ | $\mathbf{IoU_{3D} > 0.50}$ | $\mathbf{IoU_{3D} > 0.75}$ |
| living_room_traj0_frei_png | 18 | ConceptFusion (w/o rel) | 88.89 | 88.89 | 0 | 0 |
| | | ConceptFusion | 16.67 | 5.56 | 0 | 0 |
| | | OVIR-3D | 83.34 | 83.34 | 50 | 22.23 |
| | | OVSG-J | 5.56 | 5.56 | 5.56 | 0 |
| | | OVSG-S | 94.45 | 94.45 | 61.12 | 22.23 |
| | | **OVSG-L (Ours)** | **100** | **100** | **66.67** | **22.23** |
| living_room_traj1_frei_png | 34 | ConceptFusion (w/o rel) | 61.77 | 50 | 41.18 | 0 |
| | | ConceptFusion | 26.48 | 26.48 | 14.71 | 0 |
| | | OVIR-3D | 70.59 | 70.59 | 67.65 | 0 |
| | | OVSG-J | 38.24 | 38.24 | 29.42 | 0 |
| | | OVSG-S | 58.83 | 58.83 | 55.89 | 11.77 |
| | | **OVSG-L (Ours)** | **79.42** | **79.42** | **70.59** | **14.71** |
| living_room_traj2_frei_png | 28 | ConceptFusion(w/o rel) | 46.43 | 14.29 | 0 | 0 |
| | | ConceptFusion | 3.58 | 0 | 0 | 0 |
| | | OVIR-3D | 50 | 50 | 50 | 28.58 |
| | | OVSG-J | 42.86 | 35.72 | 3.58 | 0 |
| | | OVSG-S | 82.15 | 82.15 | 50 | 28.58 |
| | | **OVSG-L (Ours)** | **92.86** | **92.86** | **53.58** | **28.58** |
| living_room_traj3_frei_png | 17 | ConceptFusion(w/o rel) | 0 | 0 | 0 | 0 |
| | | ConceptFusion | 0 | 0 | 0 | 0 |
| | | OVIR-3D | 11.77 | 11.77 | 0 | 0 |
| | | OVSG-J | 23.53 | 23.53 | 23.53 | 0 |
| | | OVSG-S | 41.18 | 23.53 | 11.77 | 11.77 |
| | | **OVSG-L (Ours)** | **82.36** | **64.71** | **52.95** | **29.42** |
| office_room_traj0_frei_png | 29 | ConceptFusion(w/o rel) | 0 | 0 | 0 | 0 |
| | | ConceptFusion | 0 | 0 | 0 | 0 |
| | | OVIR-3D | 65.52 | 65.52 | 65.52 | 0 |
| | | OVSG-J | 44.83 | 44.83 | 41.38 | 0 |
| | | OVSG-S | 65.52 | 65.52 | 65.52 | 0 |
| | | **OVSG-L (Ours)** | **100** | **100** | **96.56** | **0** |
| office_room_traj1_frei_png | 19 | ConceptFusion(w/o rel) | 0 | 0 | 0 | 0 |
| | | ConceptFusion | 0 | 0 | 0 | 0 |
| | | OVIR-3D | 68.43 | 68.43 | 68.43 | 31.58 |
| | | OVSG-J | 42.11 | 42.11 | 42.11 | 21.06 |
| | | OVSG-S | 73.69 | 73.69 | 73.69 | 36.85 |
| | | **OVSG-L (Ours)** | **100** | **100** | **94.74** | **57.9** |
| office_room_traj2_frei_png | 12 | ConceptFusion(w/o rel) | 0 | 0 | 0 | 0 |
| | | ConceptFusion | 0 | 0 | 0 | 0 |
| | | OVIR-3D | 83.34 | 83.34 | 33.34 | 8.34 |
| | | OVSG-J | 0 | 0 | 0 | 0 |
| | | OVSG-S | 83.34 | 83.34 | 33.34 | 8.34 |
| | | **OVSG-L (Ours)** | **100** | **100** | **41.67** | **16.67** |
| office_room_traj3_frei_png | 25 | ConceptFusion(w/o rel) | 12 | 0 | 0 | 0 |
| | | ConceptFusion | 0 | 0 | 0 | 0 |
| | | OVIR-3D | 44 | 44 | 44 | 0 |
| | | OVSG-J | 48 | 48 | 28 | 8 |
| | | OVSG-S | 60 | 60 | 60 | 16 |
| | | **OVSG-L (Ours)** | **100** | **100** | **80** | **32** |

Table 11: Grounding Success Rate$_{3D}$ of OVSG & ConceptFusion on ICL-NUIM

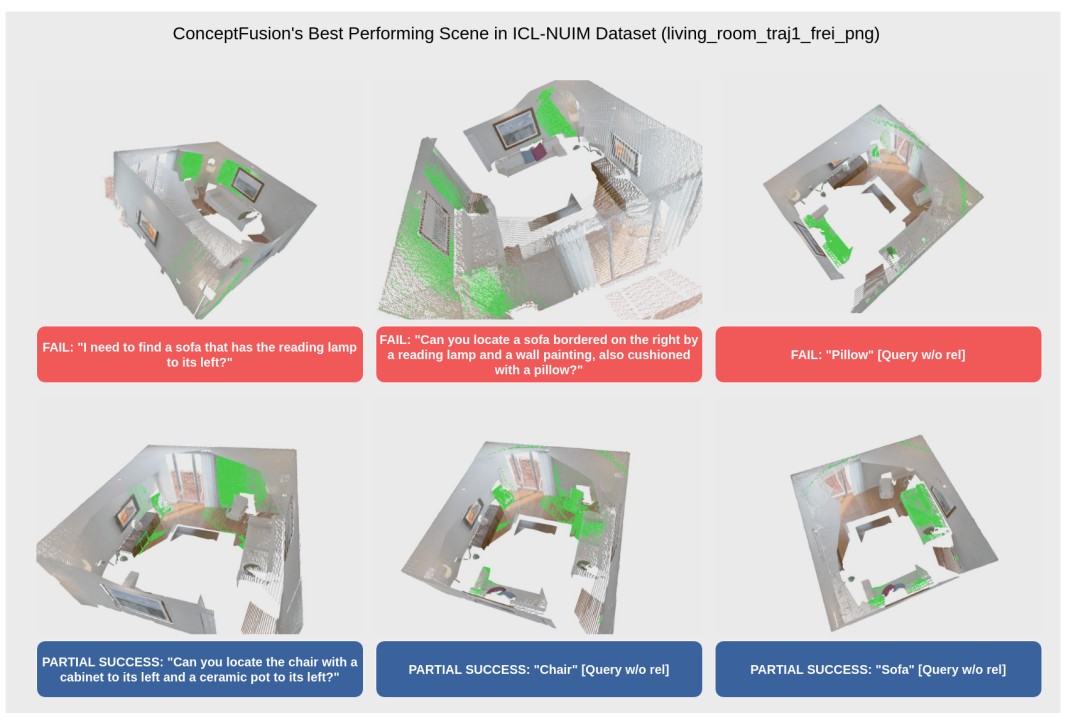

Figure 13: Performance of ConceptFusion on sample ICL-NUIM Queries

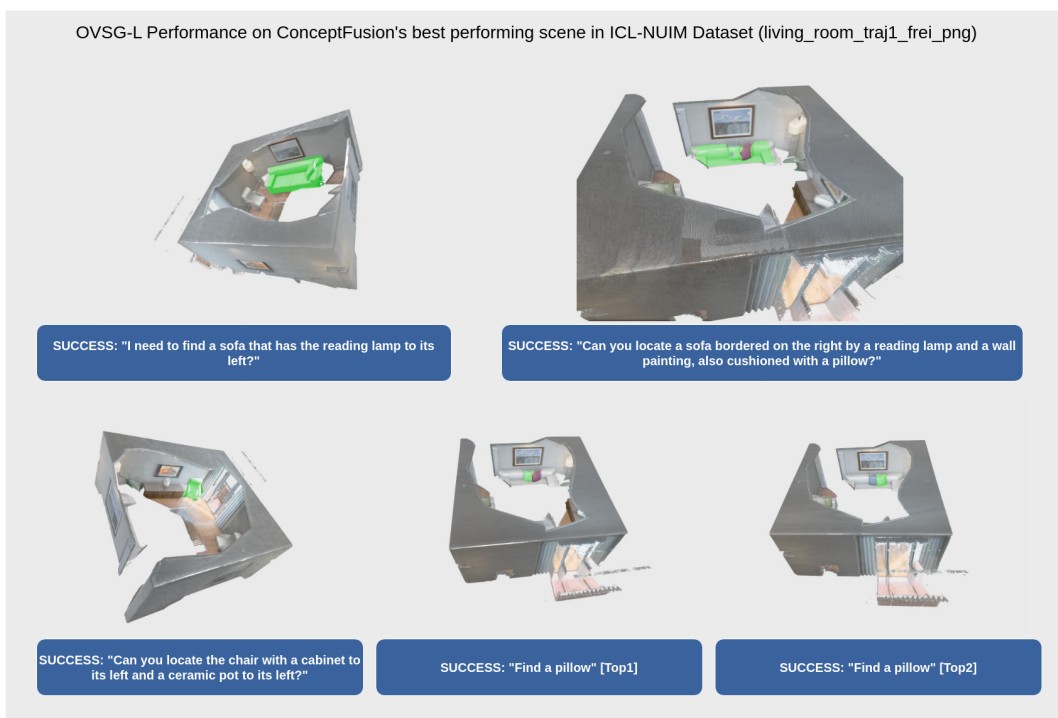

Figure 14: Performance of OVSG-L (Our method) on sample ICL-NUIM Queries

