# OpenReview forum: "Context-Aware Entity Grounding with Open-Vocabulary 3D Scene Graphs"
_robot-learning.org/CoRL/2023/Conference — CoRL 2023 Poster_

### Official Review · Reviewer_zmgd · 2023-07-01

**Confidence:** 5
**Originality:** Very Good
**Technical Quality:** Good
**Clarity Of Presentation:** Good
**Impact:** 3

**Recommendation:**

Weak Accept: I recommend accepting the paper, but will not argue for my recommendation if the majority of other reviewers have a different opinion.

**Review:**

Strengths – The main strength of this paper is the adoption of multiple language encoders to incorporate diverse linguistic expressions. \
Weaknesses – There are two major issues in this paper: clarity and effectiveness of the proposed algorithm\

This paper needs to clarify and improve a few things.\

-	The major concern is the lack of supporting evaluations. First of all, the evaluation results only show the graph matching performance and IoU which do not say the accuracy of the grounding result. Based on the provided tables, the proposed framework, OVGS-L, shows about 50-60 matching rates. However, this does not tell us the proposed framework could find correct targets. Although the matching rates are high, the grounding accuracies can be low. In that case, we cannot say the proposed method is meaningful.

-	Furthermore, another issue is the lack of baselines: this paper uses only one baseline method, OVIR-3D, that is not trained with relationships, so anyone can guess its lower performance than the proposed frameworks. For fair comparisons, this paper needs to train any methods with the relationship dataset using the dataset described in Appendix C. In addition, there must be other scene-graph-based grounding frameworks that can be used as baselines while handling relationships. We can find candidates via simple googling like [1].

[1] Kim, Dohyun, Jinwoo Kim, Minwoo Cho, and Daehyung Park. "Natural Language-Guided Semantic Navigation Using Scene Graph." In International Conference on Robot Intelligence Technology and Applications, pp. 148-156. Cham: Springer International Publishing, 2022.

-	Another biggest problem is the unclear explanation to reproduce the proposed method. First of all, the proposed framework uses LLM to parse a given instruction. Although the proposed framework largely depends on the LLM’s parsing capability, this paper does not clearly explain how the authors extracted the required information such as object, agent, region, abstract and spatial relations. Appendix C is not clear. More detail must be needed to improve the clarity of this paper and its readers.

-	Lastly, it would be better to provide deeper evaluation results and analysis about the spatial/semantic grounding accuracies instead of the open-vocabulary analysis since we know (large) language models have the capability of handling various similar expressions.

-	Figure 1 is really hard to understand. In particular, the encoding boxes are not interpretable.

-	L147-149 need to be improved for a better understanding

**Quality Of The Limitations Section:**

Limitations are addressed clearly

**Questions For Rebuttal:**

-	Why do Gs and Gq use different object feature encoders? Doesn’t it cause a bad distance estimation result?

-	In Eq. (1), the spatial and abstraction distances require scaling to balance their effects. How do you determine?

-	This paper uses different language model per feature. How do you select? Why don’t you use a specific model or all models per feature?

-	In Eq. (2), what is this \pi(k)? This is different from the explanation on L110. Further, this must be a deterministic function but equations use it as a random variable. Isn’t it mathematically incorrect?

-	L168: what is the central node and how do you select it given an instruction?

-	L169, L173: how do you extract a sub-star-graph?

-	L284-287: where is the result of the success rate?

**Robotics Focus:**

Sufficient demonstration on hardware

**Summary Of Paper:**

This paper aims to solve a problem of natural language grounding given a 3D scene graph. However, the grounding is challenging due to the variability of class category, relationship, and attribute that may be unforeseen during the construction of 3D scene graph. To figure it out, this paper proposes a context-aware entity grounding framework, an Open-Vocabulary 3D Scene Graph (OVSG), by leveraging the representation power of language models to represent unseen semantic categories and relationships. The main contributions of this paper are as follows:

-	A 3D scene graph-based grounding framework with an open-vocabulary query
-	A construction of dataset for the context-aware entity grounding
-	Demonstrations with navigation and manipulation scenarios

**Summary Of Recommendation:**

The proposed methodology is very interesting as an idea, but it is difficult to accept it without further experimental improvement due to the lack of a detailed description in the methodology and the lack of evaluation results (i.e., grounding success rates) supporting the mentioned contributions.

---

### Official Review · Reviewer_2AQt · 2023-07-20

**Confidence:** 4
**Originality:** Very Good
**Technical Quality:** Very Good
**Clarity Of Presentation:** Excellent
**Impact:** 3

**Recommendation:**

Weak Accept: I recommend accepting the paper, but will not argue for my recommendation if the majority of other reviewers have a different opinion.

**Review:**

Strengths
* Uses semantic features instead of discrete labels
* Paper is well written, notation is clear and defined, overall flow is great

Weaknesses
* Additional baselines:
  * It would be interesting to see an implementation where an open vocabulary VLM is used to capture the scene and locations of objects, and a LLM is used to parse the query and select the object of interest. How would such an implementation compare with the proposed method?
* Unclear points:
  * What is the reference frame of the position? Is it with respect to some world frame?
  * Regarding sub-graph matching, the node-level association with the highest likelihood will be selected. What happens in the scenario where all matching values are low?
  * What is the runtime and compute requirements of the method?
* Additional ablations:
  * Using different encoding mechanisms: the authors use a combination of GloVe and Sentence-BERT. What about other encoding mechanisms?
  * For sub-graph matching, the $\sigma_v$ and $\sigma_e$ are balancing parameters, what are they chosen to be? How do different values affect performance?

**Quality Of The Limitations Section:**

Limitations are addressed clearly

**Questions For Rebuttal:**

Overall, the paper was interesting to read. However, I have a few questions that I was hoping the authors could address:
* How does the method compare with a basic implementation using an open-vocabulary object detector / VLM + LLM combined approach?
* What is the reference frame for the position attribute?
* How does the method handle situations where all matching values are low?
* What is the runtime and compute requirements?
* Have the authors tried capturing the representation in the GNN? Are there preliminary results on this / what is the bottleneck?
* See review section for detailed descriptions.

**Robotics Focus:**

Sufficient demonstration on hardware

**Summary Of Paper:**

The paper introduces an Open-Vocabulary 3D Scene Graph (OVSG), a formal framework that enables context-aware entity localization in 3D scenes using free-form text queries. Unlike traditional semantic-based approaches, OVSG supports open-vocabulary querying, allowing more flexible and detailed queries. The proposed approach outperforms previous semantic-based localization techniques in comparative experiments using the ScanNet dataset and a self-collected dataset. The paper also demonstrates the practical application of OVSG in real-world robot navigation and manipulation experiments.

**Summary Of Recommendation:**

Overall, the paper was enjoyable to read. The proposed method is well described, and appears to outperform the OVIR baseline. However, I hope the authors could provide some additional baselines (i.e. a simple baseline using a combination of open-vocabulary object detector / VLM + LLM), to compare and potentially see the benefit to using a graphical approach, as opposed to using the implicit reasoning ability embedded within the LLM. For details, see review section.

---

### Official Review · Reviewer_UCiN · 2023-07-25

**Confidence:** 3
**Originality:** Very Good
**Technical Quality:** Fair
**Clarity Of Presentation:** Good
**Impact:** 3

**Recommendation:**

Weak Accept: I recommend accepting the paper, but will not argue for my recommendation if the majority of other reviewers have a different opinion.

**Review:**

I will start by summarizing what I see as the strengths and weaknesses of the paper and then follow up with some other general comments and inquiries.

Strengths:

 - The paper presents a timely method that incoporates several state-of-the-art components such as LLMs, vision language models, and 3D scene graphs.

 - The paper has a strong systems contribution where many different models and econders are all incorporated into the method which is impressive from an engineering perspective and also useful from the standpoint of the wide variability of queries that can be treated.

 - I believe this problem statement of semantic localization within a 3D scene graph is novel.


Weaknesses:

 - I would argue that the paper lacks methodological depth. The crux of the method is Eq. (1) which defines a simple distance measure over the embedding space which is used to underpin the sub-graph matching methods. I have some doubts about the regularity of these embedding spaces with respect to how well calibrated these distance measures really are. I would be interested to hear the authors' perspective on this point.

 - Further, the authors assume that the global 3D scene graph is given and that the semantic embeddings for the global scene graph are also given, so finding the matching sub-graph for the new query seems relatively straightforward (unless I misunderstood some piece of complexity)

 - I find the level of rigour of the evaluation a bit troubling. The only baseline method considered is OVIR-3D is unpublished (there is some question about how the authors know who are the authors of this work and how they got access to the unreleased code...). Further to that, the DOVE-G which is created by the authors.

 - Further, since OVIR-3D does not have access to the additional information in the form of the 3DSG, is this really a fair comparison?


Other Comments

 - I find the introduction of the manuscript a bit hard to follow. I would not consider "fiducial markers" to be a state-of-the-art method to perform natural language grounding. Other statements such as "3D scene graphs ... bridge the gap between humans and robots" are also difficult to interpret.

 - I feel that the related work should at least acknowledge the the work in language grounding that has taken place for the last decade or more before the widespread popularity of LLMs. One example from Stefanie Tellex's group (among others) is [1].

 - The authors may also consider comparing to works such as ConceptFusion [2] which builds a 3D semantic/spatial map that can be queried in the way described in this work.

 - I had difficulty to understand what is the "Positional Input" and the "User Input". Is the "Positional Input just the 3D scene graph without any semantic annotations?

 - The number of subscripts and superscripts get a little bit extreme at times. Perhaps it is unavoidable, but I found the mathematical notation somewhat laborious to follow.

[1] Grounding natural language instructions to semantic goal representations for abstraction and generalization. Dilip Arumugam, Siddharth Karamcheti, Nakul Gopalan, Edward C. Williams, Mina Rhee, Lawson L. S. Wong & Stefanie Tellex. Autonomous Robots. 2018.

[2] https://concept-fusion.github.io/

**Quality Of The Limitations Section:**

Additional details required

**Questions For Rebuttal:**

I would be interested to hear the authors' perspectives on the "Weaknesses" listed above.

**Robotics Focus:**

Relevant but unlikely to deploy to hardware in near future

**Summary Of Paper:**

The authors proposed a method of semantically localizing a query within a given 3D scene graph that is enhanced to include semantic embeddings (from various large language and vision language models). To do so they construct a small scene graph corresponding to the query and then search through the global scene graph using 3 proposed measures to quantify how well the small scene graph matches different subgraphs in the global scene graph. They demonstrate the system on synthetically generated queries over the ScanNet dataset, over manually and automatically generated queries in their own dataset, and for a robot manipulation task.

**Summary Of Recommendation:**

In summary, the authors have tackled a timely and novel problem, but I remain to be convinced that the challenges presented are very difficult to overcome or that the results have major significance based on the results they have presented.

---

### Decision · Program_Chairs · 2023-08-30

**Decision:**

Accept (Poster)

**Comment:**

This submission initially received mixed reviews.  The authors did a good job during the rebuttal phase, after which the reviewers all leaned toward acceptance.  The AC agrees with the recommendations.  Note that Reviewer UCiN pointed out that the recommendation is conditioned on the authors' promise to "release our code (including OVSG and OVIR-3D) and dataset (DOVE-G) for future robust evaluations."  The AC also agrees that the code and data release is critical given the nature of the work.  The authors should prepare code and data release along with the camera-ready version of the paper.